# Perturbation Type Categorization for Multiple Adversarial Perturbation Robustness

**Pratyush Maini**[1] **Xinyun Chen**[2] **Bo Li**[3] **Dawn Song**[2]

[1]Carnegie Mellon University
[2]University of California, Berkeley
[3]University of Illinois at Urbana-Champaign

## Abstract

Recent works in adversarial robustness have proposed defenses to improve the robustness of a single model against the union of multiple perturbation types. However, these methods still suffer significant trade-offs compared to the ones specifically trained to be robust against a single perturbation type. In this work, we introduce the problem of categorizing adversarial examples based on their perturbation types. We first theoretically show on a toy task that adversarial examples of different perturbation types constitute different distributions—making it possible to distinguish them. We support these arguments with experimental validation on multiple $\ell_p$ attacks and common corruptions. Instead of training a single classifier, we propose PROTECTOR, a two-stage pipeline that first categorizes the perturbation type of the input, and then makes the final prediction using the classifier specifically trained against the predicted perturbation type. We theoretically show that at test time the adversary faces a natural trade-off between fooling the perturbation classifier and the succeeding classifier optimized with perturbation-specific adversarial training. This makes it challenging for an adversary to plant strong attacks against the whole pipeline. Experiments on MNIST and CIFAR-10 show that PROTECTOR outperforms prior adversarial training-based defenses by over 5% when tested against the union of $\ell_1, \ell_2, \ell_\infty$ attacks. Additionally, our method extends to a more diverse attack suite, also showing large robustness gains against multiple $\ell_p$, spatial and recolor attacks.

## 1 INTRODUCTION

Machine learning models have been shown to be vulnerable to different types of adversarial examples—inputs with a small magnitude of perturbation added to mislead the classifier's prediction [Szegedy et al., 2013]. Consequently, many defenses have been proposed to improve their robustness, a majority of which focus on achieving robustness against a specific perturbation type [Goodfellow et al., 2015, Madry et al., 2018, Kurakin et al., 2017, Tramèr et al., 2018, Dong et al., 2018, Zhang et al., 2019, Carmon et al., 2019]. However, as ML models get adopted in real-world applications, it becomes important for the defenses to be robust against different types of perturbations given the flexibility of practical attackers. In addition, prior work showed that when models are trained to be robust against one perturbation type, the robustness is typically not preserved against attacks of a different type [Schott et al., 2018, Kang et al., 2019].

Motivated by the need for robustness against diverse perturbation types, recent works have attempted to train models that are robust against multiple perturbation types [Tramèr and Boneh, 2019, Maini et al., 2020, Laidlaw et al., 2021]. These works consider perturbations restricted by their $\ell_p$ norms ($p \in \{1, 2, \infty\}$) or spatial and color transformations. The proposed methods improve the overall robustness against multiple perturbation types. However, when evaluating the robustness against each individual perturbation type, the robustness of models trained by these methods is still considerably worse than those trained on a single perturbation type. Given these empirical observations, in this work we aim to answer: *Are different types of perturbations separable? Can we categorize them to improve robustness to multiple adversarial perturbations?*

To address these questions and explore the properties of different perturbation types, we introduce the problem of *categorizing adversarial examples* based on their perturbation types. We present theoretical analysis on a toy task to show that when we add different types of perturbations to benign samples of a given ground-truth class, their new dis-

*Accepted for the 38th Conference on Uncertainty in Artificial Intelligence (UAI 2022).*

tributions are distinct and separable. We experimentally validate our theoretical results on both (mathematically) well-defined perturbation regions such as $\ell_p$ balls, as well as various common corruptions [Hendrycks and Dietterich, 2019]. We find that deep networks are able to categorize different perturbation types with high accuracy ($> 95\%$). Further, our perturbation classifier shows high generalization accuracy ($\sim 90\%$) to *unseen* common corruptions, i.e., correctly predicting their categories (weather, noise, blur, or digital) without training on them. While in this work we focus on improving worst-case adversarial robustness, applications of categorizing perturbation types extend beyond it—such as detecting *systematic* distribution shifts (e.g. presence of snow for self-driving cars [Michaelis et al., 2020]). Further, using a perturbation classifier as the discriminator may improve the effectiveness and variety of adversarial examples produced by generative models [Wong and Kolter, 2021, Xiao et al., 2018a, Song et al., 2018].

Based on our theoretical analysis, we propose PROTECTOR, a two-stage pipeline that performs *Perturbation Type Categorization to Improve Robustness* against multiple perturbations. First, the top-level perturbation classifier predicts the perturbation type of the input. Then, among the second-level predictors, PROTECTOR selects the one that is the most robust to the predicted perturbation type to make the final prediction. We theoretically show that there exists a natural tension between attacking the perturbation classifier and the second-level predictors. Specifically, strong attacks against the second-level predictors make it easier for the perturbation classifier to predict the adversarial perturbation type; on the other hand, fooling the perturbation classifier requires planting weaker (or less representative) attacks against the second-level predictors. As a result, even an *imperfect* perturbation classifier significantly improves the model's overall robustness to multiple perturbation types. We also supplement our theoretical statements on the toy task with experimental validation in the exact same setting.

Empirically[1], we first show that the perturbation classifier generalizes well on classifying a wide range of adversarial perturbations. Then we compare PROTECTOR with recent defenses against multiple attack types on MNIST and CIFAR-10. Even though we do not utilize adversarial training [Goodfellow et al., 2015] to train the perturbation classifier, an ensemble of diverse perturbation classifiers along with adding small noise to inputs help make PROTECTOR robust against adaptive attacks. Specifically, we combine predictions of perturbation classifiers that classify adversarial examples in their image and Fourier domains [Yin et al., 2019a]. This further increases the tension between attacking top-level and second-level components by reducing the space of successful adversarial attacks. PROTECTOR outperforms prior approaches by over 5% against the union of

$\ell_1, \ell_2$ and $\ell_\infty$ attacks. From the suite of 15 different attacks tested, the average improvement over all the attacks w.r.t. the state-of-art baseline defense is $\sim 15\%$ on both MNIST and CIFAR-10. Training a model to be robust against multiple attacks typically imposes a significant tradeoff against the accuracy on benign samples, but PROTECTOR attains $\sim 7\%$ greater benign test accuracy on CIFAR-10 as compared to recent works [Laidlaw et al., 2021, Maini et al., 2020]. We further demonstrate how our defense naturally extends beyond $\ell_p$ perturbation types, where we assess the robustness of our model against the union of $\ell_\infty$, $\ell_2$, spatial [Wong et al., 2019, Xiao et al., 2018b] and recolor [Bhattad et al., 2020, Laidlaw and Feizi, 2019] attacks on CIFAR-10. Our defense exceeds the robustness of recent work [Laidlaw et al., 2021] by over 13% against all attacks. In addition, PROTECTOR provides the flexibility to plug in and integrate new defenses against individual perturbation types into the existing framework as second-level predictors, thus the defense performance of PROTECTOR can be continuously improved with the development of more advanced defenses against single perturbation types.

## 2 RELATED WORK

**Adversarial examples.** Among the different types of adversarial attacks studied in prior work [Szegedy et al., 2013, Goodfellow et al., 2015, Madry et al., 2018, Hendrycks et al., 2019, Bhattad et al., 2020], the majority constrain the perturbation within a small $\ell_p$ region around the original input. To improve model robustness in the presence of such adversaries, most existing defenses utilize adversarial training [Goodfellow et al., 2015], which augments the training dataset with adversarial examples. Till date, different variants of adversarial training algorithms remain the most successful defenses against adversarial attacks [Carmon et al., 2019, Zhang et al., 2019, Wong et al., 2020, Rice et al., 2020, Wang et al., 2020]. Other types of defenses include input transformation [Guo et al., 2018, Buckman et al., 2018] and network distillation [Papernot et al., 2016], but were rendered ineffective under stronger adversaries [He et al., 2017, Carlini and Wagner, 2017a, Athalye et al., 2018, Tramer et al., 2020].

**Defenses against multiple perturbation types.** Some recent works have focused on defending against a union of norm bounded $\ell_p$ attacks. Schott et al. [2018], Kang et al. [2019] showed that models that were trained for a given $\ell_p$-norm bounded attack are not robust against attacks in a different $\ell_q$ region. Schott et al. [2018] proposed the use of multiple variational autoencoders to achieve robustness to multiple $\ell_p$ attacks on MNIST. Tramèr and Boneh [2019] used simple aggregations of multiple adversaries to achieve non-trivial robust accuracy against $\ell_1, \ell_2, \ell_\infty$ attacks. Maini et al. [2020] proposed MSD that takes gradient steps in the union of multiple $\ell_p$ regions to improve multiple perturbation robustness. Most recently, Laidlaw et al. [2021] pro-

---

[1]Code for reproducing our experiments can be found at https://github.com/sunblaze-ucb/adversarial-protector.

posed a defense against unseen perturbations using perceptual adversarial training. They evaluate their work against $\ell_\infty$, $\ell_2$, spatial, recolor adversaries.

**Detection of adversarial examples.** Multiple prior works have focused on detecting adversarial examples [Feinman et al., 2017, Lee et al., 2018, Ma et al., 2018, Cennamo et al., 2020, Fidel et al., 2019, Yin et al., 2019b]. However, most of these methods were rendered ineffective in the presence of adaptive adversaries [Carlini and Wagner, 2017a, Tramer et al., 2020]. In comparison, our work focuses on a more challenging problem of categorizing perturbation types. To this end, Yin et al. [2019a] proposed the examination of Fourier transforms of adversarial examples to determine the adversarial attack and corruption types.

# 3 SEPARABILITY OF PERTURBATION TYPES

In this section, we formally illustrate the setup of perturbation categorization. In Theorem 1, we show the existence of a classifier that can separate adversarial examples belonging to different perturbation types. We focus on $\ell_p$ attacks (that can be fully specified mathematically) on a simplified binary classification task for the convenience of theoretical analysis. However, PROTECTOR can also improve the empirical robustness of models trained on common image classification benchmarks against both $\ell_p$ and non-$\ell_p$ attacks. We will discuss the empirical examination in Section 6.

## 3.1 PROBLEM SETTING

**Data distribution.** We consider a distribution $\mathcal{D}$ of inputs sampled from the union of two multi-variate Gaussian distributions such that the input-label pairs $(x, y)$ can be described as:

$$y \overset{u.a.r}{\sim} \{-1, +1\},$$
$$x_0 \sim \mathcal{N}(y\alpha, \sigma^2), \quad x_1, \ldots, x_d \overset{i.i.d}{\sim} \mathcal{N}(y\eta, \sigma^2), \quad (1)$$

where $x = [x_0, x_1, \ldots, x_d] \in \mathcal{R}^{d+1}$ and $\eta = \frac{\alpha}{\sqrt{d}}$. This setting demonstrates the distinction between a feature $x_0$ that is strongly correlated with the label, and $d$ weakly correlated features that are (independently) normally distributed with the mean $y\eta$ and the variance $\sigma^2$. In our work, we assume that $\frac{\alpha}{\sigma} > 10$ ($x_0$ is strongly correlated) and $d > 100$ (remaining $d$ features are weakly correlated, but together represent a strongly correlated feature). This setting was adapted from Ilyas et al. [2019], and more discussion can be found in Appendix A.

**Perturbation types.** We focus our theoretical discussion on adversaries constrained within a fixed $\ell_p$ region of radius $\epsilon_p$ around the original input, for $\ell_p \in \mathcal{S} = \{\ell_1, \ell_\infty\}$. Such adversaries are frequently studied in existing work for finding

the optimal first-order perturbation for different attack types. Let $\ell(\cdot, \cdot)$ be the cross-entropy loss, and $\Delta_{\mathcal{S}} = \bigcup_{\ell_p \in \mathcal{S}} \Delta_{\ell_p, \epsilon}$ for the $\ell_p$ threat model, $\Delta_{\ell_p, \epsilon_p}$, of radius $\epsilon_p$. Then, for a model $f_\theta$, the optimal perturbation $\delta^*$ is given by:

$$\delta^* = \arg \max_{\delta \in \Delta_{\mathcal{S}}} \ell(f_\theta(x + \delta), y). \quad (2)$$

## 3.2 SEPARABILITY OF $\ell_p$ PERTURBATIONS

Consider a classifier $M$ trained with the objective of correctly classifying inputs $x \in \mathcal{D}$. The goal of the adversary is to fool $M$ by finding the optimal perturbation $\delta_{\mathcal{A}} \ \forall \mathcal{A} \in S$. The theorem below shows that the distributions of adversarial inputs within different $\ell_p$ regions can be separated with a high accuracy.

**Theorem 1** (Separability of perturbation types). *Given a binary Gaussian classifier $M$ trained on $\mathcal{D}$, consider $\mathcal{D}_p^y$ to be the distribution of optimal adversarial inputs (for a class $y$) against $M$, within $\ell_p$ regions of radius $\epsilon_p$, where $\epsilon_1 = \alpha$, $\epsilon_\infty = \alpha/\sqrt{d}$. Distributions $\mathcal{D}_p^y$ ($p \in \{1, \infty\}$) can be accurately separated by a binary Gaussian classifier $C_{adv}$ with a misclassification probability $P_e \leq 10^{-24}$.*

The proof sketch is as follows. We first calculate the optimal weights of a binary Gaussian classifier $M$ trained on $\mathcal{D}$. Accordingly, for any input $x \in \mathcal{D}$, we find the optimal adversarial perturbation $\delta_{\mathcal{A}} \ \forall \mathcal{A} \in \{\ell_1, \ell_\infty\}$ against $M$. We discuss how these perturbed inputs $x + \delta_{\mathcal{A}}$ also follow a normal distribution, with shifted means. Finally, for data points of a given label, we show that $C_{adv}$ is able to predict the correct perturbation type with a very low error. We present the formal proof in Appendix B.

# 4 PROTECTOR: PERTURBATION TYPE CATEGORIZATION FOR ROBUSTNESS

We illustrate the PROTECTOR pipeline in Figure 1. PROTECTOR performs the classification task as a two-stage process. Given an input, PROTECTOR first utilizes a *perturbation classifier* $C_{adv}$ to predict its perturbation type. Then, based on the predicted type, PROTECTOR uses the corresponding second-level predictor $M_{\mathcal{A}}$ to provide the final prediction, where $M_{\mathcal{A}}$ is specially trained to be robust against the attack $\mathcal{A} \in S$. Formally, let $f_\theta$ be the PROTECTOR model, then:

$$f_\theta(x) = M_{\mathcal{A}}(x); \quad s.t. \quad \mathcal{A} = \arg\max C_{adv}(x). \quad (3)$$

## 4.1 ADVERSARIAL TRADE-OFF

In Section 3.2, we showed that the optimal perturbations of different attack types belong to different data distributions, and can be separated by a simple classifier. However, in the white-box setting, the adversary has knowledge of both the

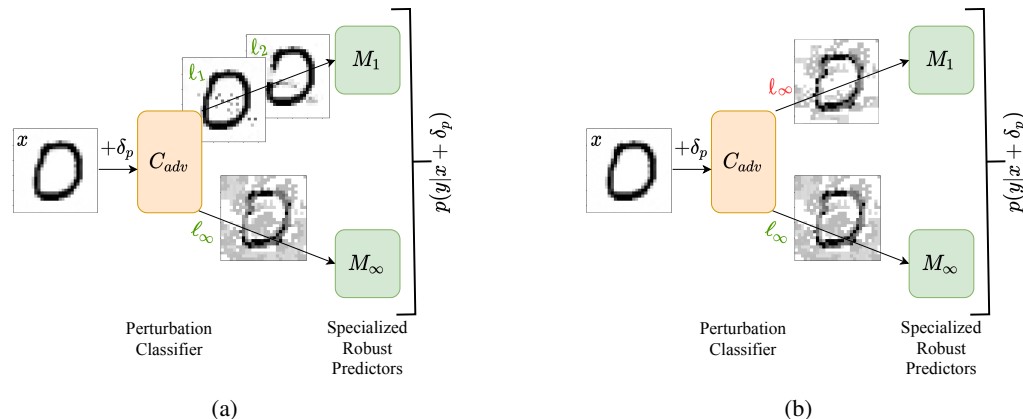

Figure 1: An overview of PROTECTOR. (a) The perturbation classifier $C_{adv}$ categorizes representative attacks of different types. (b) An illustration of the trade-off in Theorem 2. An adversarial example fooling $C_{adv}$ (the $\ell_\infty$ sample marked in red) becomes weaker to attack the second-level $M_{\mathcal{A}}$ models. Stronger or more representative attacks (marked green) are correctly categorized.

perturbation classifier ($C_{adv}$) and specialized robust models ($M_{\mathcal{A}}$). This allows it to adapt the attack to fool the entire pipeline instead of individual models alone. To validate the robustness of PROTECTOR, we provide a theoretical justification in Theorem 2, showing that PROTECTOR naturally offers a trade-off between fooling $C_{adv}$ and the individual models $M_{\mathcal{A}}$. This makes it difficult for adversaries to stage successful attacks against PROTECTOR.

Note that there are some overlapping regions among different perturbation constraints. For example, every adversary could set $\delta_p = 0$ as a valid perturbation, in which case $C_{adv}$ can not correctly classify all attacks. However, such perturbations are not useful to the adversary, because any $M_{\mathcal{A}}$ can correctly classify unperturbed inputs with a high probability. In the following theorem, we examine the robustness of PROTECTOR in the presence of such strong dynamic adversaries.

**Theorem 2** (Adversarial trade-off). *Given a data distribution $\mathcal{D}$, adversarially trained models $M_{\ell_p,\epsilon_p}$, and an attack classifier $C_{adv}$ that distinguishes perturbations of different $\ell_p$ attack types for $p \in \{1, \infty\}$; the probability of a successful attack by the strongest adversary over the* PROTECTOR *pipeline is $P_e < 0.01$ for $\epsilon_1 = \alpha + 2\sigma$ and $\epsilon_\infty = \frac{\alpha + 2\sigma}{\sqrt{d}}$.*

Here, the *worst-case adversary* refers to an adaptive adversary that has full knowledge of the defense strategy. In Appendix C.2, we discuss how $\epsilon_1, \epsilon_\infty$ are set so that the $\ell_1$ and $\ell_\infty$ adversaries can fool $M_{\ell_\infty,\epsilon_\infty}$ and $M_{\ell_1,\epsilon_1}$ models respectively with a high success rate. To prove Theorem 2, we first show that when trained on $\mathcal{D}$, an adversarially robust model $M_{\mathcal{A}}$ can achieve robust accuracy $> 99\%$ against the attack type it was trained for, and $< 2\%$ against an alternate attack. By "alternate" we mean that for an $\ell_q$ attack, the prediction is made by the $M_{\ell_p,\epsilon_p}$ model. Then, we analyze the modified distributions of the inputs perturbed by different $\ell_p$ attacks. Based on this, we construct a simple decision rule for the perturbation classifier $C_{adv}$. Finally, we compute the

perturbation induced by the worst-case adversary. We show that there exists a trade-off between fooling the $C_{adv}$ (to allow the alternate $M_{\ell_p,\epsilon_p}$ model to make the final prediction for an $\ell_q$ attack $\forall p, q \in \{1, \infty\}; p \neq q$), and fooling the alternate $M_{\ell_p,\epsilon_p}$ model itself. We provide an illustration of the trade-off in Figure 4b, and a formal proof and *experimental validation* on the toy task in Appendix C.

## 5 TRAINING AND INFERENCE

We now extend PROTECTOR to deep neural networks trained on common image classification benchmarks. Following prior work on defending against multiple perturbation types, we evaluate on MNIST [LeCun et al., 2010] and CIFAR-10 [Krizhevsky, 2012] datasets. Here, we present the training details, the formulation of an ensemble of perturbation classifiers, and adaptive white-box attacks against PROTECTOR.

### 5.1 DATASET CREATION

To train our perturbation classifier $C_{adv}$, we create a dataset that includes adversarial examples of different perturbation types. We perform adversarial attacks against each of the individual $M_{\mathcal{A}}$ models used in PROTECTOR to curate the training and test sets. In the case of $\ell_p$ examples, we use the PGD attack [Madry et al., 2018], and for spatial [Xiao et al., 2018b] and recolor [Laidlaw and Feizi, 2019] attacks, we use their original attack formulation. The time for creating the dataset against each $M_{\mathcal{A}}$ is the same as running a single epoch of adversarial training. Since most recent works typically train their models for ~200 epochs, the dataset creation time is insignificant when compared with the cost of training an $M_{\mathcal{A}}$ model.

**Combining perturbation types.** When training PROTECTOR to be robust against a set $\mathcal{S}$ of multiple ($k$) attacks, we

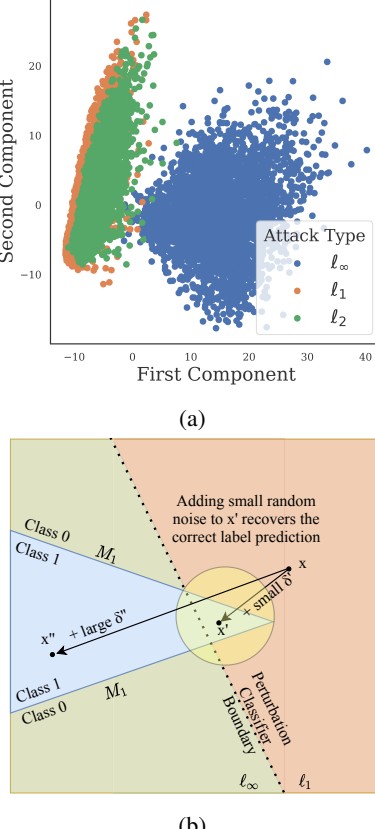

(a)

(b)

Figure 2: (a) PCA for different adversarial perturbations on MNIST. (b) Illustration of the effect of random noise on generating adversarial examples. The notion of small, large perturbations is only used to illustrate the scenario in Figure 2b, and neither perturbation region subsumes the other.

combine certain perturbation types under the same label to improve the overall robustness. This is beneficial when: (a) a specialized model $M_{\mathcal{A}}$ also shows a high degree of robustness to a different attack $\mathcal{B} \in \mathcal{S}$, s.t. $\mathcal{A} \neq \mathcal{B}$; (b) two different attack types $\mathcal{A}, \mathcal{B} \in \mathcal{S}$ have similar characteristics. For instance, in case of $\ell_p$ attacks, we perform binary classification between $\mathcal{A} = \{\{\ell_1, \ell_2\}, \ell_\infty\}$. We hypothesize that compared to $\ell_\infty$ adversarial examples, $\ell_1$ and $\ell_2$ adversarial examples show similar characteristics. To provide an intuitive illustration, we randomly sample 10K adversarial examples generated with PGD attacks on MNIST, and present their Principal Component Analysis (PCA) in Figure 2a. We observe that the first two principal components for $\ell_1$ and $\ell_2$ adversarial examples are largely overlapping, while those for $\ell_\infty$ are clearly from a different distribution.[2] For the MNIST dataset, we use the $M_{\ell_2}, M_{\ell_\infty}$ models in PROTECTOR, and we use $M_{\ell_1}, M_{\ell_\infty}$ models for CIFAR-10. The choice is made based on the robustness of $\{M_{\ell_2}, M_{\ell_1}\}$ models against $\{\ell_1, \ell_2\}$ attacks respectively, as will be depicted in Table 2. Similarly, when defending against the union of

$\ell_p$ and non-$\ell_p$ perturbation types on CIFAR-10, we classify $\mathcal{A} = \{\{\ell_\infty, \ell_2, \text{ReColor}\}, \text{StAdv}\}$ attacks based on the robustness of each $M_{\mathcal{A}}$ against every attack $\mathcal{B} \in \mathcal{S}$. We report the robustness of PROTECTOR with varying number of second-level predictors in Appendix J.3.

## 5.2 TRAINING

Past works [Maini et al., 2020, Tramèr and Boneh, 2019] on robustness to multiple attack types require intensive hyperparameter tuning to *balance* different attack types when one attack is stronger than others. We find that a similar phenomenon plagues the adversarial training (AT) of $C_{adv}$. Therefore, we train $C_{adv}$ over a static dataset, which is fast and stable. Specifically, using a single GTX 1080Ti GPU, $C_{adv}$ can be trained within 5 and 30 minutes on MNIST and CIFAR-10 respectively (given that we already have access to perturbation-specific robust models). On the other hand, training state-of-the-art models robust to a single perturbation type requires up to 2 days to train on the same amount of GPU power, and existing defenses against multiple ($k$) perturbation types take $k$ times as long as the training time for robustness against a single perturbation type. Instead, even when the individual $M_{\mathcal{A}}$ are unavailable, we can train the $k$ models in parallel to improve training speed.

A key advantage of PROTECTOR's design is that it can build upon existing defenses against individual perturbation types. Specifically, we leverage the adversarially trained models developed in prior work [Zhang et al., 2019, Carmon et al., 2019] as $M_{\mathcal{A}}$ models in our pipeline. The architecture of $C_{adv}$ is also similar to a single $M_{\mathcal{A}}$ model. See Appendix D for more details.

## 5.3 INFERENCE PROCEDURE

**Ensemble of diverse perturbation classifiers.** While $C_{adv}$ learns the ability to distinguish between different attack types, it is not immune to the presence of adaptive adversaries that try to fool $C_{adv}$ and the $M_{\mathcal{A}}$ models together. To improve model robustness against such adversaries, we attempt to increase the trade-off in PROTECTOR that was described in Section 4.1. We use an ensemble (average of prediction logits) of two perturbation classifiers that classify adversarial examples in different domains – via the Fourier and image domains.[3] Owing to this diversity, the classification landscape of each $C_{adv}$ is different. Intuitively, the trade-off between fooling the two stages of PROTECTOR confines the adversary in a very small region for generating successful adversarial attacks when using an ensemble of perturbation classifiers. In Appendix G, we show how the adversarial examples can be visually separated in the

---

[2]The visualization only serves as motivation. It does not suggest that $\ell_1, \ell_2$ examples are not separable.

[3]Adversaries can still back-propagate through the Fourier transformation steps.

Fourier domain [Yin et al., 2019a] and discuss further implementation details of the ensemble.

**Constraining the adversary using random noise.** While past work has [Hu et al., 2019] suggested that adding random noise does not help defend against adversarial inputs, it is the *unique* exhibition of the trade-off described in Theorem 2 that adversarial attacks against PROTECTOR, on the contrary, are likely to fail when added with random noise. Intuitively, the trade-off between fooling the two stages of PROTECTOR confines the adversary in a very small region for crafting successful attacks.

Consider the illustrative example in Figure 2b. The input $(x, y = 0)$ is subjected to an $\ell_\infty$ attack. Assume that the $M_{\ell_\infty, \epsilon_\infty}$ model is a perfect classifier for adversarial examples within a fixed $\epsilon_\infty$ region. The dotted line shows the decision boundary for $C_{adv}$, which correctly classifies inputs subjected to $\ell_\infty$ perturbations $\delta''$ as $\ell_\infty$ attacks (green), but misclassifies samples with smaller perturbations. When the adversary adds a large perturbation $\delta''$, the prediction of $M_{\ell_1}$ for the resulted input $x''$ becomes wrong, but the perturbation classifier also categorizes it as an $M_{\ell_\infty}$ attack, thus the final prediction of PROTECTOR is still correct since it will be produced by $M_{\infty, \epsilon_\infty}$ model instead. On the other hand, when the adversary adds a small perturbation $\delta'$ to fool the perturbation classifier, adding a small amount of random noise can recover the correct prediction with a high probability. Note that every point on the boundary of the noise region (yellow circle) is correctly classified by the pipeline. In this way, adding random noise exploits an adversarial trade-off for PROTECTOR to achieve a high accuracy against adversarial examples, in the absence of adversarial training. In our implementation, we sample random noise $z \sim \mathcal{N}(0, I)$, and add $\hat{z} = \epsilon_2 \cdot z/|z|_2$ to the model input.

### 5.4 ADAPTIVE ATTACKS AGAINST PROTECTOR

**Gradient propagation.** Since the final prediction in Equation 3 only depends on a single $M_\mathcal{A}$ model, the pipeline does not allow gradient flow across the two levels. This can make it difficult for gradient-based adversaries to attack PROTECTOR. Therefore, we utilize a combination of predictions from each individual $M_\mathcal{A}$ model by modifying $f_\theta(x)$ in Equation 3 as follows:

$$c = \text{softmax}(C_{adv}(x));$$
$$f_\theta(x) = \sum_{\mathcal{A} \in \mathcal{S}} c_\mathcal{A} \cdot M_\mathcal{A}(x), \qquad (4)$$

where $c_\mathcal{A}$ denotes the probability of the input $x$ being classified as the perturbation type $\mathcal{A}$ by $C_{adv}$. Equation 4 is only used for the purpose of generating adversarial examples and performing gradient-based attack optimization. For consistency, we still use Equation 3 to compute the model prediction at inference (final forward-propagation). We do not see any significant performance advantages of either

choice during inference, and briefly report a comparison in Appendix I.1.

**Separately attacking $C_{adv}$ and $M_\mathcal{A}$.** We also experiment with other strategies of aggregating the predictions of different components, e.g., tuning the loss to balance direct attacks on $C_{adv}$ and each $M_\mathcal{A}$ model. We find that this attack formulation performs worse than attacking the entire pipeline with Equation 4. We provide a discussion on this attack in Appendix I.

## 6 EXPERIMENTS

In this section, we present our results on MNIST and CIFAR-10 datasets, both for the perturbation classifier $C_{adv}$ alone, and for the entire PROTECTOR pipeline.

### 6.1 PERTURBATION CATEGORIZATION BY $C_{adv}$

**Categorizing $\ell_p$ perturbations.** First, we justify our choice of $\epsilon_p$ radii by empirically quantifying the overlapping regions of different types of adversarial attacks. We observe that the empirical overlap is exactly 0% in all cases on both MNIST and CIFAR-10, and we present the full analysis in Appendix H.1. We then evaluate the categorization performance of $C_{adv}$ on a dataset of adversarial examples which are generated against the six models we use as the baseline defenses in our experiments. Note that $C_{adv}$ is only trained on adversarial examples against the two $M_\mathcal{A}$ models that are part of PROTECTOR.

Next, we evaluate the test set generalization across the various datasets created. We observe that $C_{adv}$ transfers well across the board. First, $C_{adv}$ generalizes to adversarial examples against new models, i.e., it preserves a high accuracy, even if the adversarial examples are generated against models that are unseen during training. Further, $C_{adv}$ also generalizes to new attack algorithms. As discussed in Section 5.1, we only include PGD adversarial examples in our training set for $C_{adv}$. However, on adversarial examples generated by the AutoAttack library, the classification accuracy of $C_{adv}$ still holds up. In particular, the accuracy is $> 95\%$ across all the individual test sets created. These results suggest two important findings that validate our results in Theorem 1 — independent of **(a)** the model to be attacked; and **(b)** the algorithm for generating the optimal adversarial perturbation, the optimal adversarial images for a given $\ell_p$ region follow similar distributions. We present the full results in Appendix H.2.

**Categorizing common corruptions.** CIFAR-10-C is a benchmark consisting of 19 different types of common corruptions [Hendrycks and Dietterich, 2019]. For each image in the original CIFAR-10 test set, CIFAR-10-C includes images with different corruptions. To train the corruption classifier, we split CIFAR-10-C, so that each corruption type has 9K training samples, and 1K for testing. For corruptions

Table 1: Generalization results when $C_{adv}$ is trained on different Noise, Blur, Weather and Digital corruptions (Severity=5). Test is performed on Speckle Noise + Gaussian Blur + Spatter + Saturate.

| Trained On | Accuracy |
|---|---|
| Impulse + Defocus Blur + Snow + Brightness | 70.4% |
| + Gaussian + Glass Blur + Fog + Contrast | 80.1% |
| + Shot + Motion Blur + Frost + Elastic Trans | 85.6% |
| + Zoom Blur + JPEG Compression + Pixelate | 93.5% |
| + Speckle + Gaussian Blur + Spatter + Saturate | 99.8% |

of the highest severity, we observe that our corruption classifier achieves greater than 99% test accuracy on the test split. Details about the architecture are deferred to Appendix D. This demonstrates that our perturbation classifier is applicable to both $\ell_p$ adversarial perturbations and semantic common corruptions. We discuss detailed results of corruption classification at various severity levels in Appendix H.3.

**Generalization to unseen corruptions.** We further evaluate the generalization of the perturbation classifier to unseen corruption types. Specifically, different from the above setting of classifying corruption types, now our classifier categorizes all corruption types into 4 categories — noise, blur, digital, and weather (as defined in the CIFAR-10-C benchmark). We evaluate the model performance on 4 held-out corruption types, 1 for each category, and select these corruption types following the model validation setting in Hendrycks and Dietterich [2019]. From the remaining 15 corruption types, we vary the number of corruptions included for training, and present the results in Table 1. We observe that even if we do not train the perturbation classifier on the same corruption types for testing, the classifier still obtains a high generalization accuracy ($> 90\%$). These results demonstrate that perturbation classification is effective even for unseen perturbations.

## 6.2 ROBUSTNESS TO $\ell_p$ ATTACKS

**Baselines.** We compare PROTECTOR with the state-of-art defenses against the union of $\ell_1, \ell_2, \ell_\infty$ adversaries. For Tramèr and Boneh [2019], we compare two variants of adversarial training: (1) the **MAX** approach, where for each image, among different perturbation types, the adversarial sample that leads to the maximum increase of the model loss is augmented into the training set; (2) the **AVG** approach, where adversarial examples for all perturbation types are included for training. We also compare with **MSD** [Maini et al., 2020], which modifies the standard PGD attack to incorporate the union of multiple perturbation types within the steepest decent. In addition, we evaluate $\mathbf{M}_{\ell_1}, \mathbf{M}_{\ell_2}, \mathbf{M}_{\ell_\infty}$ models trained with $\ell_1, \ell_2, \ell_\infty$ perturbations separately, as described in Appendix D.

**Attack evaluation.** We evaluate against the strongest attacks in the adversarial examples literature, and with adaptive attacks specifically designed for PROTECTOR (Section 5.4). We perform standard PGD attacks along with attacks from the AutoAttack library [Croce and Hein, 2020b], which achieves the state-of-art adversarial error rates against multiple recently published models. The radius of the $\{\ell_1, \ell_2, \ell_\infty\}$ perturbation regions is $\{10, 2, 0.3\}$ for the MNIST dataset and $\{10, 0.5, 0.03\}$ for the CIFAR-10 dataset. We present the full details of attack algorithms in Appendix F.

Following prior work, we evaluate models on adversarial examples generated from the first 1000 images of the test set for MNIST and CIFAR-10. Our main evaluation metric is the accuracy on *all attacks* – a given input is a failure case if any of the attack algorithm in our suite successfully fools the model.

**Results.** In Table 2, we summarize the worst-case performance against all attacks of a given perturbation type for MNIST and CIFAR-10 datasets. In particular, "Ours" denotes the robustness of PROTECTOR against the adaptive attacks described in Section 5.4, and "Ours*" denotes the robustness of PROTECTOR against standard attacks based on Equation 3. The adaptive strategy effectively reduces the overall accuracy of PROTECTOR by $2 - 5\%$, showing that incorporating the gradient and prediction information of all second-level predictors results in a stronger attack.

PROTECTOR outperforms all baselines by $6.4\%$ on MNIST, and $10\%$ on CIFAR-10 in terms of the *all attacks* metric, even when evaluated against a strong adaptive adversary. Compared to the previous state-of-art defense against multiple perturbation types (MSD), the accuracy gain on $\ell_\infty$ attacks is especially notable, i.e., around $15\%$. In particular, if we compare the performance on each individual attack algorithm, as shown in Appendix J.1 and J.2 for MNIST and CIFAR-10 respectively, the average accuracy gain is $\sim 15\%$ for both datasets. These results demonstrate that PROTECTOR considerably mitigates the trade-off in the accuracy for individual attacks. Further, PROTECTOR retains a $7\%$ higher CIFAR-10 accuracy on *clean images*, as opposed to past defenses that sacrifice benign accuracy for robustness to multiple perturbation types.

## 6.3 ROBUSTNESS TO NON-$\ell_p$ ATTACKS

We demonstrate how PROTECTOR can be extended to perturbation types beyond those restricted to $\ell_p$ types. Laidlaw et al. [2021] evaluate the robustness of various adversarial defenses against attacks $\mathcal{A} \in \mathcal{S} = \{\ell_2, \ell_\infty, \text{StAdv}, \text{ReColor}\}$ on CIFAR-10. We directly compare PROTECTOR with the pre-trained models for each individual defense provided in their work. This includes their defense based on perceptual adversarial training (**PAT**) and the **MAX**, **AVG** models, along with perturbation-

Table 2: Worst-case accuracies against different $\ell_p$ attacks: (a) MNIST; (b) CIFAR-10. *Ours* represents PROTECTOR against the adaptive attack strategy (Eq 4), and *Ours\** is the standard setting.

| MNIST | $M_{\ell_\infty}$ | $M_{\ell_2}$ | $M_{\ell_1}$ | MAX | AVG | MSD | Ours | Ours* |
|---|---|---|---|---|---|---|---|---|
| Clean accuracy | 99.2% | 98.7% | 98.8% | 98.6% | 99.1% | 98.3% | 98.9% | 98.9% |
| $\ell_\infty$ attacks ($\epsilon = 0.3$) | 90.2% | 2.6% | 0.0% | 39.0% | 57.8% | 63.5% | 78.1% | 79.0% |
| $\ell_2$ attacks ($\epsilon = 2.0$) | 9.5% | 72.3% | 47.8% | 58.5% | 58.6% | 65.7% | 66.6% | 72.3% |
| $\ell_1$ attacks ($\epsilon = 10$) | 18.8% | 70.6% | 77.5% | 41.8% | 46.1% | 64.3% | 68.1% | 72.5% |
| All attacks | 7.3% | 2.6% | 0.0% | 29.1% | 37.1% | 57.2% | **63.6%** | **67.2%** |

(a)

| CIFAR-10 | $M_{\ell_\infty}$ | $M_{\ell_2}$ | $M_{\ell_1}$ | MAX | AVG | MSD | Ours | Ours* |
|---|---|---|---|---|---|---|---|---|
| Clean accuracy | 89.5% | 93.9% | 89.0% | 81.0% | 84.6% | 81.7% | 89.0% | 89.0% |
| $\ell_\infty$ attacks ($\epsilon = 0.03$) | 59.3% | 34.8% | 35.0% | 34.9% | 39.7% | 43.7% | 56.1% | 58.4% |
| $\ell_2$ attacks ($\epsilon = 0.5$) | 64.6% | 77.2% | 71.5% | 61.8% | 65.5% | 64.5% | 69.3% | 69.4% |
| $\ell_1$ attacks ($\epsilon = 10$) | 27.6% | 45.3% | 60.9% | 43.7% | 60.0% | 56.1% | 57.9% | 59.5% |
| All attacks | 27.6% | 32.9% | 35.0% | 31.5% | 39.3% | 43.5% | **53.5%** | **54.9%** |

(b)

Table 3: Worst-case accuracies against $\ell_\infty$ ($\epsilon = 0.003$), $\ell_2$ ($\epsilon = 0.5$), spatial and recolor attacks. *Ours* represents PROTECTOR against the adaptive attack strategy (Eq 4), and *Ours\** is the standard setting. PAT [Laidlaw et al., 2021] is trained using perceptual adversarial training.

| CIFAR-10 | $M_{\ell_\infty}$ | $M_{\ell_2}$ | $M_{\text{StAdv}}$ | $M_{\text{ReColor}}$ | MAX | AVG | PAT | Ours | Ours* |
|---|---|---|---|---|---|---|---|---|---|
| Clean acc. | 89.5% | 93.9% | 86.2% | 93.4% | 84.0% | 86.8% | 71.6% | 89.5% | 89.5% |
| $\ell_\infty$ attacks | 59.3% | 34.8% | 0.1% | 8.5% | 25.8% | 42.1% | 29.8% | 58.2% | 59.1% |
| $\ell_2$ attacks | 64.6% | 77.2% | 10.0% | 34.8% | 44.2% | 64.8% | 54.1% | 57.0% | 57.2% |
| StAdv | 5.7% | 0.2% | 68.9% | 0.0% | 46.2% | 27.8% | 58.4% | 50.4% | 55.7% |
| ReColor | 85.5% | 84.0% | 52.1% | 86.8% | 77.4% | 80.5% | 70.9% | 85.2% | 85.3% |
| All attacks | 5.4% | 0.2% | 0.1% | 0.0% | 24.0% | 21.5% | 27.8% | **40.9%** | **41.9%** |

specific robust models $M_A$. Specifically, as discussed in Section 5.1, we train a perturbation classifier that classifies adversarial examples as belonging to one of the two classes: $\{\{\ell_\infty, \ell_2, \text{ReColor}\}, \text{StAdv}\}$. We use two individual robust predictors: $\{M_{\ell_\infty}, M_{\text{StAdv}}\}$. The choice is once again made based on the robust accuracy of $M_{\ell_\infty}$ models against $\{\ell_\infty, \ell_2, \text{ReColor}\}$ attacks as also presented in Table 3. This ability to combine attacks also represents positively on the scalability of PROTECTOR. PROTECTOR improves by 13.1% against the union of all attacks. Importantly, PROTECTOR preserves a high accuracy against benign samples, whereas PAT classifies only 71.6% of unperturbed samples correctly, which makes it difficult to adopt it in real-world settings.

## 7 CONCLUSION

In this work, we introduce the problem of categorizing perturbation types. We theoretically demonstrate that adversarial inputs of different attack types are separable, and empirically validate our claims on different $\ell_p$ and non-$\ell_p$ attacks. In addition to categorizing them with high accuracy, the perturbation categorizer also generalizes to *unseen* corruptions of the same category.

PROTECTOR performs perturbation type categorization to achieve robustness against the union of multiple perturbation types. We theoretically examine the existence of a natural tension for any adversary trying to fool our model—between fooling the attack classifier and the specialized robust predictors. Our empirical results on MNIST and CIFAR-10 datasets complement our theoretical analysis, showing that PROTECTOR outperforms existing defenses against multiple $\ell_p$ and non-$\ell_p$ attacks by over 5%, while showing gains of over $\sim 15\%$ on average and clean accuracy metrics.

Our work serves as a stepping stone towards the goal of universal adversarial robustness, by dissecting multiple adversarial objectives into individually solvable pieces and combining them via PROTECTOR. In its present form, PROTECTOR requires the knowledge of each individual attack type that we want to be robust against—to train the perturbation classifier. This limitation opens up various avenues for future work, including the new problem of perturbation categorization by defining sub-classes of adversarial attack types, and training generative models to synthesize diverse perturbations.

## Acknowledgements

This material is in part based upon work supported by the National Science Foundation under Grant No. TWC-1409915, Berkeley DeepDrive, and DARPA D3M under Grant No. FA8750-17-2-0091. Any opinions, findings, and conclusions or recommendations expressed in this material are those of the author(s) and do not necessarily reflect the views of the National Science Foundation. Xinyun Chen is supported by the Facebook Fellowship.

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

## A    PROBLEM SETTING: THEORETICAL ANALYSIS

The classification problem consists of two tasks: **(1)** Predicting the correct class label of an adversarially perturbed (or benign) image using adversarially robust classifier $M_{\mathcal{A}}$; and **(2)** Predicting the type of adversarial perturbation that the input image was subjected to, using attack classifier $C_{adv}$.

**Setup.**    We consider the data to consist of inputs to be sampled from two multi-variate Gaussian distributions such that the input-label pairs (x,y) can be described as:

$$
y \overset{u.a.r}{\sim} \{-1, +1\}, \\
x_0 \sim \mathcal{N}(y\alpha, \sigma^2), \quad x_1, \ldots, x_d \overset{i.i.d}{\sim} \mathcal{N}(y\eta, \sigma^2), \tag{5}
$$

where the input $x \sim \mathcal{N}(y\boldsymbol{\mu}, \boldsymbol{\Sigma}) \in \mathcal{R}^{(d+1)}$; $\eta = \alpha/\sqrt{d}$ for some positive constant $\alpha$; $\boldsymbol{\mu} = [\alpha, \eta, \ldots, \eta] \in \mathcal{R}^{+(d+1)}$ and $\boldsymbol{\Sigma} = \sigma^2 \mathbf{I} \in \mathcal{R}^{+(d+1)\times(d+1)}$. We can assume without loss of generality, that the mean for the two distributions has the same absolute value, since for any two distributions with mean $\boldsymbol{\mu}_1, \boldsymbol{\mu}_2$, we can translate the origin to $\frac{\boldsymbol{\mu}_1 + \boldsymbol{\mu}_2}{2}$. This setting demonstrates the distinction between an input feature $x_0$ that is strongly correlated with the input label and $d$ weakly correlated features that are normally distributed (independently) with mean $y\eta$ and variance $\sigma^2$ each. We adapt this setting from Ilyas et al. [2019] who used a stochastic feature $x_0 = y$ with probability $p$, as opposed to a normally distributed input feature as in our case. All our findings hold in the other setting as well, however, the chosen setting better represents true data distribution, with some features that are strongly correlated to the input label, while others that have only a weak correlation.

## B    SEPARABILITY OF PERTURBATION TYPES (THEOREM 1)

Our goal is to evaluate if the optimal perturbation confined within different $\ell_p$ balls have different distributions and whether they are separable. We do so by developing an error bound on the maximum error in classification of the perturbation types. The goal of the adversary is to fool a standard (non-robust) classifier $M$. $C_{adv}$ aims to predict the perturbation type based on **only** viewing the adversarial image, and not the delta perturbation.

First, in Appendix B.1 we define a binary Gaussian classifier that is trained on the given task. Given the weights of the binary classifier, we then identify the optimal adversarial perturbation for each of the $\ell_1, \ell_2, \ell_\infty$ attack types in Appendix B.2. In Appendix B.3 we define the difference between the adversarial input distribution for different $\ell_p$ balls. Finally, we calculate the error in classification of these adversarial input types in Appendix B.4 to conclude the proof of Theorem 1.

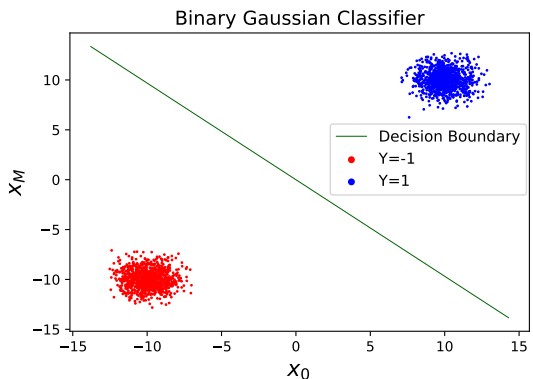

Figure 3: **Simulation**: Decision boundary (solid green line) of binary Gaussian classifier. $x_M = \frac{1}{\sqrt{d}} \sum_{i=1}^{d} x_i$ represents a meta feature, and $x_0$ is the first dimension of the input.

### B.1    BINARY GAUSSIAN CLASSIFIER

We assume that we have enough input data to be able to empirically estimate the parameters $\mu, \sigma$ of the input distribution via sustained sampling. The multivariate Gaussian representing the input data is given by:

$$p(x|y = y_i) = \frac{1}{\sqrt{(2\pi)^d|\Sigma|}} \exp\left(-\frac{1}{2}(x - y_i.\boldsymbol{\mu})^T \Sigma^{-1}(x - y_i.\boldsymbol{\mu})\right), \quad \forall y_i \in \{-1, 1\}. \tag{6}$$

We want to find $p(y = y_i|x) \, \forall y_i \in \{-1, +1\}$. From Bayesian Decision Theory, the optimal decision rule for separating the two distributions is given by:

$$p(y = 1)p(x|y = 1) \overset{y=1}{>} p(y = -1)p(x|y = -1);$$
$$p(y = 1)p(x|y = 1) \overset{y=-1}{<} p(y = -1)p(x|y = -1). \tag{7}$$

Therefore, for two Gaussian Distributions $\mathcal{N}(\boldsymbol{\mu}_1, \boldsymbol{\Sigma}_1), \mathcal{N}(\boldsymbol{\mu}_2, \boldsymbol{\Sigma}_2)$, we have:

$$0 \overset{y=1}{<} x^\top A x - 2b^\top x + c;$$
$$A = \boldsymbol{\Sigma}_1^{-1} - \boldsymbol{\Sigma}_2^{-1};$$
$$b = \boldsymbol{\Sigma}_1^{-1}\mu_1 - \boldsymbol{\Sigma}_2^{-1}\mu_2;$$
$$c = \mu_1^\top \boldsymbol{\Sigma}_1^{-1}\mu_1 - \mu_2^\top \boldsymbol{\Sigma}_2^{-1}\mu_2 + \log \frac{\|\Sigma_1\|}{\|\Sigma_2\|} - 2\log \frac{p(y = 1)}{p(y = -1)}. \tag{8}$$

Substituting (6) and (7) in (8), we find that the optimal Bayesian decision rule for our problem is given by:

$$x^\top \boldsymbol{\mu} \overset{y=1}{>} 0, \tag{9}$$

which means that the label for the input can be predicted with the information of the sign of $x^\top \boldsymbol{\mu}$ alone. We can define the parameters $\mathbf{W} \in \mathcal{R}^{d+1}$ of the optimal binary Gaussian classifier $M^W$, such that $\|\mathbf{W}\|_2 = 1$ as:

$$\mathbf{W}_0 = \frac{\alpha}{\sqrt{2}}, \qquad \mathbf{W}_i = \frac{\alpha}{\sqrt{2d}} \quad \forall i \in \{1, \ldots, d\};$$
$$M^W(x) = x^\top W. \tag{10}$$

The same is also verified via a simulation in Figure 3.

## B.2 OPTIMAL ADVERSARIAL PERTURBATION AGAINST $M^W$

Now, we calculate the optimal perturbation $\delta$ that is added to an input by an adversary in order to fool our model. For the purpose of this analysis, we only aim to fool a model trained on the standard classification metric as discussed in Section 3 (and not an adversarially robust model). The parameters of our model are defined in (10).

The objective of any adversary $\delta \in \Delta$ is to maximize the loss of the label classifier $M^W$. We assume that the classification loss is given by $-y \times M^W(x + \delta)$. The object of the adversary is to find $\delta^*$ such that:

$$\ell(x + \delta, y; M^W) = -y \times M^W(x + \delta) = -yx^\top \mathbf{W};$$
$$\delta^* = \arg\max_{\delta \in \Delta} \ell(x + \delta, y; M^W),$$
$$= \arg\max_{\delta \in \Delta} -y(x + \delta)^\top \mathbf{W} = \arg\max_{\delta \in \Delta} -y\delta^\top \mathbf{W}. \tag{11}$$

We will now calculate the optimal perturbation in the $\ell_p$ balls $\forall p \in \{1, 2, \infty\}$. For the following analyses, we restrict the perturbation region $\Delta$ to the corresponding $\ell_p$ ball of radius $\{\epsilon_1, \epsilon_2, \epsilon_\infty\}$ respectively. We also note that the optimal perturbation exists at the boundary of the respective $\ell_p$ balls. Therefore, the constraint can be re-written as :

$$\delta^* = \arg \max_{\|\delta\|_p = \epsilon_p} -y\delta^\top \mathbf{W}. \tag{12}$$

We use the following properties in the individual treatment of $\ell_p$ balls:

$$\|\delta\|_p = \left(\sum_i |\delta_i|^p\right)^{\frac{1}{p}},$$

$$\partial_j \|\delta\|_p = \frac{1}{p}\left(\sum_i |\delta_i|^p\right)^{\frac{1}{p}-1} \cdot p|\delta_j|^{p-1} \operatorname{sgn}(\delta_j) = \left(\frac{|\delta_j|}{\|\delta\|_p}\right)^{p-1} \operatorname{sgn}(\delta_j). \tag{13}$$

**p = 2** Making use of langrange multipliers to solve (12), we have:

$$\nabla_\delta(-\delta^\top \Sigma^{-1}\mu) = \lambda\nabla_\delta(\|\delta\|_p^2 - \epsilon_p^2),$$

$$-\mathbf{W} = \lambda'\|\delta\|_p\nabla_\delta(\|\delta\|_p). \tag{14}$$

Combining the results from (13) and replacing $\delta$ with $\delta_2$ we obtain :

$$-\mathbf{W} = \lambda'\|\delta_2\|_2\left(\frac{|\delta_2|}{\|\delta_2\|_2}\right)\operatorname{sgn}(\delta_2)$$

$$\delta_2; = -\epsilon_2\left(\frac{\mathbf{W}}{\|\mathbf{W}\|_2}\right) = -\epsilon_2\mathbf{W}. \tag{15}$$

**p = ∞** Recall that the optimal perturbation is given by :

$$\delta^* = \arg \max_{\|\delta\|_\infty = \epsilon_\infty} -y\delta^\top \mathbf{W},$$

$$= \arg \max_{\|\delta\|_\infty = \epsilon_\infty} -y\sum_{i=0}^d \delta_i \mathbf{W}_i. \tag{16}$$

Since $\|\delta\|_\infty = \epsilon_\infty$, we know that $\max_i |\delta_i| = \epsilon_\infty$. Therefore (16) is maximized when each $\delta_i = -y\epsilon_\infty \operatorname{sgn} \mathbf{W}_i \quad \forall i \in \{0, \dots, d\}$. Further, since the weight matrix only contains non-negative elements ($\alpha$ is a positive constant), we can conclude that the optimal perturbation is given by:

$$\delta_\infty = -y\epsilon_\infty \mathbf{1}. \tag{17}$$

**p = 1** We attempt an analytical solution for the optimal perturbation $\delta_1$. Recall that the optimal perturbation is given by :

$$\delta^* = \arg \max_{\|\delta\|_1 = \epsilon_1} -y\sum_{i=1}^d \delta_i \mathbf{W}_i,$$

$$= \arg \max_{\|\delta\|_1 = \epsilon_1} -y\delta_0 \mathbf{W}_0 - y\sum_{i=1}^d \delta_i \mathbf{W}_i, \tag{18}$$

$$= \arg \max_{\|\delta\|_1 = \epsilon_1} -y\delta_0 \frac{\alpha}{\sqrt{2}} - y\sum_{i=1}^d \delta_i \frac{\alpha}{\sqrt{2d}}.$$

Since $\|\delta\|_1 = \epsilon_1$, (18) is maximized when:

$$\delta_0 = -y\epsilon_1 \operatorname{sgn}(\alpha) = -y\epsilon_1, \qquad \delta_i = 0 \quad \forall i \in \{1 \dots d\}. \tag{19}$$

**Combining the results.**   From the preceding discussion, it may be noted that the new distribution of inputs within a given label changes by a different amount $\delta$ depending on the perturbation type. Moreover, if the mean and variance of the distribution of a given label are known (which implies that the corresponding true data label is also known), the optimal perturbation is independent of the input itself, and only dependent on the respective class statistics (Note that the input is still important in order to understand the true class).

## B.3   PERTURBATION CLASSIFICATION BY $C_{adv}$

Now we aim to verify if it is possible to accurately separate the optimal adversarial inputs crafted within different $\ell_p$ balls. For the purposes of this discussion, we only consider the problem of classifying perturbation types into $\ell_1$ and $\ell_\infty$, but the same analysis may also be extended more generally to any number of perturbation types.

We will consider the problem of classifying the correct attack label for inputs from true class $y = 1$ for this discussion. Note that the original distribution:

$$X_{true} \sim \mathcal{N}(y.\boldsymbol{\mu},\, \boldsymbol{\Sigma}).$$

Since the perturbation value $\delta_p$ is fixed for all inputs corresponding to a particular label, the new distribution of perturbed inputs $X_1$ and $X_\infty$ in case of $\ell_1$ and $\ell_\infty$ attacks respectively (for y = 1) is given by:

$$
\begin{aligned}
X_1 &\sim \mathcal{N}(\boldsymbol{\mu} + \delta_1,\, \boldsymbol{\Sigma}); \\
X_\infty &\sim \mathcal{N}(\boldsymbol{\mu} + \delta_\infty,\, \boldsymbol{\Sigma}).
\end{aligned}
\tag{20}
$$

We now try to evaluate the conditions under which we can separate the two Gaussian distributions with an acceptable worst-case error.

## B.4   CALCULATING A BOUND ON THE ERROR

**Classification Error.**   A classification error occurs if a data vector x belongs to one class but falls in the decision region of the other class. That is in (7) the decision rule indicates the incorrect class. (This can be understood through the existence of outliers)

$$
\begin{aligned}
P_e &= \int P(\text{error}|x)p(x)dx, \\
&= \int \min\left[p(y = \ell_1|x)p(x), p(y = \ell_\infty|x)p(x)\right]dx.
\end{aligned}
\tag{21}
$$

**Perturbation Size.**   We set the radius of the $\ell_\infty$ ball, $\epsilon_\infty = \eta$ and the radius of the $\ell_1$ ball, $\epsilon_1 = \alpha$. We further extend the discussion about suitable perturbation sizes in Appendix C.2. These values ensure that the $\ell_\infty$ adversary can make all the weakly correlated labels meaningless by changing the expected value of the adversarial input to less than 0 ($\mathbf{E}[x_i + \delta_\infty(i)] \quad \forall i > 0$), while the $\ell_1$ adversary can make the strongly correlated feature $x_0$ meaningless by changing its expected value to less than 0 ($\mathbf{E}[x_0 + \delta_1(0)]$). However, neither of the two adversaries can flip all the features together.

**Translating the axes.**   We can translate the axis of reference by $\left(-\mu - \left(\frac{\delta_1 + \delta_\infty}{2}\right)\right)$ and define $\boldsymbol{\mu}_{adv} = \left(\frac{\delta_1 - \delta_\infty}{2}\right)$, such that :

$$
\begin{aligned}
X_1 &\sim \mathcal{N}(\boldsymbol{\mu}_{adv},\, \boldsymbol{\Sigma}); \\
X_\infty &\sim \mathcal{N}(-\boldsymbol{\mu}_{adv},\, \boldsymbol{\Sigma}).
\end{aligned}
\tag{22}
$$

We can once again combine this with the simplified Bayesian model in (9) to obtain the classification rule:

$$
x^\top \boldsymbol{\mu}_{adv} \overset{p=1}{>} 0.
\tag{23}
$$

Combining the optimal perturbation definitions in (17) and (19) that $\boldsymbol{\mu}_{adv} = \left(\frac{\delta_1 - \delta_\infty}{2}\right) = \frac{1}{2}\left[-\epsilon_1 + \epsilon_\infty, \epsilon_\infty, \ldots, \epsilon_\infty\right]$. We can further substitute $\epsilon_1 = \alpha$ and $\epsilon_\infty = \eta = \frac{\alpha}{\sqrt{d}}$. Notice that $\boldsymbol{\mu}_{adv}(i) > 0 \;\forall i > 0$. Without loss of generality, to

simplify further discussion we can flip the coordinates of $x_0$, since all dimensions are independent of each other. Therefore, $\boldsymbol{\mu}_{adv} = \frac{\alpha}{2\sqrt{d}} \left[ \sqrt{d} - 1, 1, \ldots, 1 \right]$. Consider a new variable $x_z$ such that:

$$x_z = x_0 \cdot \left( 1 - \frac{1}{\sqrt{d}} \right) + \frac{1}{\sqrt{d}} \sum_{i=1}^{d} x_i = \frac{2}{\alpha} \left( x^\top \boldsymbol{\mu}_{adv} \right). \tag{24}$$

Since each $x_i \forall i \geq 0$ is independently distributed, the new feature $x_z \sim \mathcal{N}(\mu_z, \sigma_z^2)$, where

$$\mu_z = \alpha \left( 1 - \frac{1}{\sqrt{d}} \right) + \frac{1}{\sqrt{d}} \sum_{i=1}^{d} \frac{\alpha}{\sqrt{d}} = 2\alpha - \frac{\alpha}{\sqrt{d}}$$

$$\sigma_z^2 = \sigma^2 \left( 1 + \frac{1}{d} - 2\frac{1}{\sqrt{d}} + \sum_{i=1}^{d} \frac{1}{d} \right), \tag{25}$$

$$= \sigma^2 \left( 2 + \frac{1}{d} - 2\frac{1}{\sqrt{d}} \right).$$

Therefore, the problem simplifies to calculating the probability that the meta-variable $x_z > 0$.

For $\frac{\alpha}{\sigma} > 10$ and $d > 1$, we have in the z-table, $z > 10$:

$$P_e \leq 10^{-24}, \tag{26}$$

which suggests that the distributions are significantly distinct and can be easily separated. This concludes the proof for Theorem 1.

**Note:** We can extend the analysis to other $\ell_p$ balls as well, but we consider $\ell_1$ and $\ell_\infty$ for simplicity.

## C   ROBUSTNESS OF THE PROTECTOR PIPELINE (THEOREM 2)

In the previous section, we show that it is indeed possible to distinguish between the distribution of inputs of a given class that were subjected to $\ell_1$ and $\ell_\infty$ perturbations over a standard classifier. Now, we aim to develop further understanding of the robustness of our two-stage pipeline in a dynamic attack setting with multiple labels to distinguish among. The first stage is a preliminary classifier $C_{adv}$ that classifies the perturbation type and the second stage consists of multiple models $M_{\mathcal{A}}$ that were specifically trained to be robust to perturbations to the input within the corresponding $\ell_p$ norm.

First, in Appendix C.1, we calculate the optimal weights for a binary Gaussian classifier $M_{\mathcal{A}}$, trained on dataset $\mathcal{D}$ to be robust to adversaries within the $\ell_p$ ball $\forall p \in \{1, \infty\}$. Based on the weights of the individual model, we fix the perturbation size $\epsilon_p$ to be only as large, as is required to fool the alternate model with high probability. Here, by 'alternate' we mean that for an $\ell_q$ attack, the prediction should be made by the $M_{\ell_p, \epsilon_p}$ model, where $p, q \in \{1, \infty\}; p \neq q$. In Appendix C.3 we calculate the robustness of individual $M_{\mathcal{A}}$ models to $\ell_p$ adversaries, given the perturbation size $\epsilon_p$ as defined in Appendix C.2. In Appendix C.4, we analyze the modified distributions of the perturbed inputs after different $\ell_p$ attacks. Based on this analysis, we construct a simple decision rule for the perturbation classifier $C_{adv}$. Finally, in Appendix C.5 we determine the perturbation induced by the worst-case adversary that has complete knowledge of both $C_{adv}$ and $M_{\ell_p, \epsilon_p} \forall p \in \{1, \infty\}$. We show how there exists a trade-off between fooling the perturbation classifier (to allow the alternate $M_{\ell_p, \epsilon_p}$ model to make the final prediction), and fooling the alternate $M_{\ell_p, \epsilon_p}$ model itself.

**Perturbation Size.**   We set the radius of the $\ell_\infty$ ball, $\epsilon_\infty = \eta + \zeta_\infty$ and the radius of the $\ell_1$ ball, $\epsilon_1 = \alpha + \zeta_1$, where $\zeta_p$ are some small positive constants that we calculate in Appendix C.2. These values ensure that the $\ell_\infty$ adversary can make all the weakly correlated labels meaningless by changing the expected value of the adversarial input to less than 0 ($\mathbf{E}[x_i + \delta_\infty(i)] \quad \forall i > 0$), while the $\ell_1$ adversary can make the strongly correlated feature $x_0$ meaningless by changing its expected value to less than 0 ($\mathbf{E}[x_0 + \delta_1(0)]$). However, neither of the two adversaries can flip all the features together. The exact values of $\zeta_p$ determine the exact success probability of the attacks. We defer this calculation to later when we have calculated the weights of the models $M_{\mathcal{A}}$. For the following discussion, it may be assumed that $\zeta_p \to 0 \forall p \in \{1, \infty\}$.

## C.1 BINARY GAUSSIAN CLASSIFIER $M_{\mathcal{A}}$

Extending the discussion in Appendix B.1, we now examine the learned weights of a binary Gaussian classifier $M_{\mathcal{A}}$ that is trained to be robust against perturbations within the corresponding $\ell_p$ ball of radius $\epsilon_p$. The optimization equation for the classifier can be formulated as follows:

$$\min_{\mathbf{W}} \mathbb{E}\left[-yx^\top \mathbf{W}\right] + \frac{1}{2}\lambda\|\mathbf{W}\|_2^2, \tag{27}$$

where $\lambda$ is tuned in order to make the $\ell_2$ norm of the optimal weight distribution, $\|\mathbf{W}^*\|_2, = 1$. Following the symmetry argument in Lemma D.1 [Tsipras et al., 2018] we extend for the binary Gaussian classifier that :

$$\mathbf{W}_i^* = \mathbf{W}_j^* = \mathbf{W_M} \quad \forall i, j \in \{1, \ldots, d\}. \tag{28}$$

We deal with the cases pertaining to $p \in \{\infty, 1\}$ in this section. For both the cases, we consider existential solutions for the classifier $M_{\mathcal{A}}$ to simplify the discussion. This gives us lower bounds on the performance of the optimal robust classifier. The robust objective under adversarial training can be defined as:

$$\min_{\mathbf{W}} \max_{\|\delta\|_p \leq \epsilon_p} \mathbb{E}\left[\mathbf{W}_0 \cdot (x_0 + \delta_0) + \mathbf{W_M} \cdot \sum_{i=1}^d (x_i + \delta_i)\right] + \frac{1}{2}\lambda\|\mathbf{W}\|_2^2;$$
$$\min_{\mathbf{W}} \left\{-1\left(\mathbf{W}_0\alpha + d \times \mathbf{W_M}\frac{\alpha}{\sqrt{d}}\right) + \frac{1}{2}\lambda\|\mathbf{W}\|_2^2 + \max_{\|\delta\|_p \leq \epsilon_p} \mathbb{E}\left[-y\left(\mathbf{W}_0\delta_0 + \mathbf{W_M}\sum_{i=1}^d \delta_i\right)\right]\right\} \tag{29}$$

Further, since the $\lambda$ constraint only ensures that $\|\mathbf{W}^*\|_2 = 1$, we can simplify the optimization equation by substituting $\mathbf{W_0} = \sqrt{1 - d \cdot \mathbf{W_M}^2}$ as follows,

$$\min_{\mathbf{W_M}} \left\{-1\left(\alpha\sqrt{1 - d \cdot \mathbf{W_M}^2} + d \times \mathbf{W_M}\frac{\alpha}{\sqrt{d}}\right) + \max_{\|\delta\|_p \leq \epsilon_p} \mathbb{E}\left[-y\left(\delta_0\sqrt{1 - d \cdot \mathbf{W_M}^2} + \mathbf{W_M}\sum_{i=1}^d \delta_i\right)\right]\right\}. \tag{30}$$

**p = $\infty$**  As discussed in (17) the optimal perturbation $\delta_\infty$ is given by $-y\epsilon_\infty \mathbf{1}$. The optimization equation is simplified to:

$$\min_{\mathbf{W_M}} \left\{(\epsilon_\infty - \alpha)\sqrt{1 - d \cdot \mathbf{W_M}^2} + d \times \mathbf{W_M}\left(\epsilon_\infty - \frac{\alpha}{\sqrt{d}}\right)\right\}. \tag{31}$$

Recall that $\epsilon_\infty = \frac{\alpha}{\sqrt{d}} + \zeta_\infty$. To simplify the following discussion we use the weights of a classifier trained to be robust against perturbations within the $\ell_\infty$ ball of radius $\epsilon_\infty = \frac{\alpha}{\sqrt{d}}$. The optimal solution is then given by:

$$\lim_{\zeta_\infty \to 0} \mathbf{W_M} = 0. \tag{32}$$

Therefore, the classifier weights are given by $\mathbf{W} = [\mathbf{W}_0, \mathbf{W}_1, \ldots, \mathbf{W}_d] = [1, 0, \ldots, 0]$. We also show later in Appendix C.3 that the model achieves greater than 99% accuracy against $\ell_\infty$ adversaries for the chosen values of $\zeta_\infty$.

**p = 1**  We consider an analytical solution to yield optimal weights for this case. Recall from (19) that the optimal perturbation $\delta_1$ depends on the weight distribution of the classifier. Therefore, if $\mathbf{W}_0 > \mathbf{W_M}$ the optimization equation can be simplified to

$$\min_{\mathbf{W}} \left\{\mathbf{W_0}(\epsilon_1 - \alpha) - d \times \mathbf{W_M}\frac{\alpha}{\sqrt{d}} + \frac{1}{2}\lambda\|\mathbf{W}\|_2^2\right\}, \tag{33}$$

and if $\mathbf{W_M} > \mathbf{W}_0$

$$\min_{\mathbf{W}} \left\{-\mathbf{W}_0\alpha - \mathbf{W_M}\left(\sqrt{d}\alpha - \epsilon_1\right) + \frac{1}{2}\lambda\|\mathbf{W}\|_2^2\right\}. \tag{34}$$

Recall that $\epsilon_1 = \alpha + \zeta_1$. Once again to simplify the discussion that follows we will lower bound the robust accuracy of the classifier $M_{\ell_1}$ by considering the optimal solution when $zeta_1 = 0$. The optimal solution is then given by:

$$\lim_{\zeta_1 \to 0} \mathbf{W_M} = 1. \tag{35}$$

For the robust classifier $M_{\ell_1}$, the weights $\mathbf{W} = [\mathbf{W}_0, \mathbf{W}_1, \ldots, \mathbf{W}_d] = [0, \frac{1}{\sqrt{d}}, \frac{1}{\sqrt{d}}, \ldots, \frac{1}{\sqrt{d}}]$. While this may not be the optimal solution for all values of $\zeta_1$, we are only interested in a lower bound on the final accuracy and the classifier described by weights $\mathbf{W}$ simplifies the discussion hereon. We also show later in Appendix C.3 that the model achieves greater than 99% accuracy against $\ell_1$ adversaries for the chosen values of $\zeta_1$.

## C.2 PERTURBATION SIZES FOR FOOLING $M_{\mathcal{A}}$ MODELS

Now that we exactly know the weights of the learned robust classifiers $M_{\ell_1}$ and $M_{\ell_\infty}$, we can move towards calculating values $\zeta_1$ and $\zeta_\infty$ for the exact radius of the perturbation regions for the $\ell_1$ and $\ell_\infty$ metrics. We set the radii of these regions in such a way that an $\ell_1$ adversary can fool the model $M_{\ell_\infty}$ with probability $\sim 98\%$ (corresponding to $z = 2$ in the z-table for normal distributions), and similarly, the success of $\ell_\infty$ attacks against the $M_{\ell_1}$ model is $\sim 98\%$.

Let $P_{p_1, p_2}$ represent the probability that model $M_{\ell_{p_1}}$ correctly classifies an adversarial input in the $\ell_{p_2}$ region. For $p_1 = \infty$ and $p_2 = 1$,

$$\begin{aligned}
P_{\infty, 1} &= \mathbb{P}_{x \sim \mathcal{N}(y\boldsymbol{\mu}, \boldsymbol{\Sigma})}[y \cdot M_{\ell_\infty}(x + \delta_1) > 0], \\
&= \mathbb{P}_{x \sim \mathcal{N}(y\boldsymbol{\mu}, \boldsymbol{\Sigma})}[y \cdot (x + \delta_1)^\top \mathbf{W} > 0], \\
&\geq \mathbb{P}_{x \sim \mathcal{N}(\boldsymbol{\mu}, \boldsymbol{\Sigma})}[x_0 > \epsilon_1]; \\
z &= \frac{\epsilon_1 - \alpha}{\sigma} = \frac{\alpha + \zeta_1 - \alpha}{\sigma} = \frac{\zeta_1}{\sigma} = 2; \\
\zeta_1 &= 2\sigma; \\
\epsilon_1 &= \alpha + 2\sigma.
\end{aligned} \tag{36}$$

To simplify the discussion for the $M_{\ell_1}$ model, we define a meta-feature $x_M$ as:

$$x_M = \frac{1}{\sqrt{d}} \sum_{i=1}^{d} x_i, \tag{37}$$

which is distributed as :

$$x_M \sim \mathcal{N}(y\eta\sqrt{d}, \sigma^2) \stackrel{d}{=} \mathcal{N}(y\alpha, \sigma^2).$$

For $p_1 = 1$ and $p_2 = \infty$,

$$\begin{aligned}
P_{1, \infty} &= \mathbb{P}_{x \sim \mathcal{N}(y\boldsymbol{\mu}, \boldsymbol{\Sigma})}[y \cdot M_{\ell_1}(x + \delta_\infty) > 0], \\
&= \mathbb{P}_{x \sim \mathcal{N}(y\boldsymbol{\mu}, \boldsymbol{\Sigma})}[y \cdot (x + \delta_\infty)^\top \mathbf{W} > 0], \\
&= \mathbb{P}_{x \sim \mathcal{N}(y\boldsymbol{\mu}, \boldsymbol{\Sigma})}\left[y \cdot \frac{1}{\sqrt{d}} \sum_{i=1}^{d}(x_i + \delta_\infty(i)) > 0\right], \\
&= \mathbb{P}_{x \sim \mathcal{N}(y\boldsymbol{\mu}, \boldsymbol{\Sigma})}[y \cdot (x_M - \sqrt{d} \cdot \epsilon_\infty) > 0], \\
&\geq \mathbb{P}_{x \sim \mathcal{N}(\boldsymbol{\mu}, \boldsymbol{\Sigma})}\left[x_M > \sqrt{d} \cdot \epsilon_\infty\right]; \\
z &= \frac{\sqrt{d} \cdot \epsilon_\infty - \alpha}{\sigma} = \frac{\alpha + \sqrt{d} \cdot \zeta_\infty - \alpha}{\sigma} = \frac{\sqrt{d} \cdot \zeta_\infty}{\sigma} = 2; \\
\zeta_\infty &= \frac{2\sigma}{\sqrt{d}}; \\
\epsilon_\infty &= \frac{\alpha + 2\sigma}{\sqrt{d}};
\end{aligned} \tag{38}$$

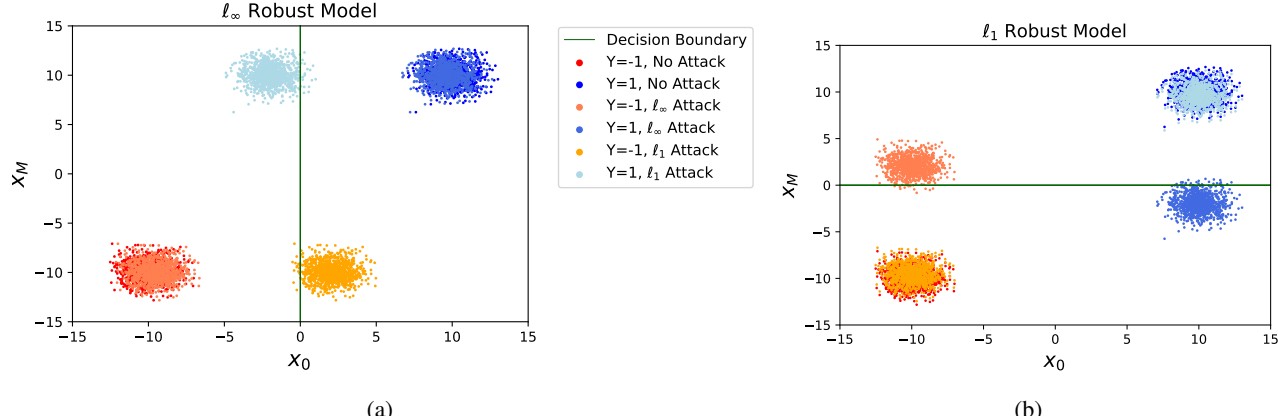

(a)                      (b)

Figure 4: **Simulation:** Decision boundary (solid green line) and robustness of individual $M_\mathcal{A}$ models to different $\ell_p$ attacks. $x_M$ represents the meta feature as defined in Equation 37 and $x_0$ is the first dimension of the input. Notice how the distribution of perturbed samples varies according to the change in model architecture (scatter plots in the same color in the two graphs represent the same distribution). (a) The $M_{\ell_\infty}$ model is able to correctly classify all benign and $\ell_\infty$ perturbed samples. However, the $\ell_1$ adversary is able to successfully flip the decision of most data points (b) The same illustration is repeated for the $M_1$ model. In this case, while the model is robust to $\ell_1$ attacks, it fails against an $\ell_\infty$ adversary.

## C.3   ROBUSTNESS OF INDIVIDUAL $M_\mathcal{A}$ MODELS

**Additional assumptions.** We add the following assumptions: (1) the dimensionality parameter $d$ of input data is larger than 100; and (2) the ratio of the mean and variance for feature $x_0$ is greater than 10. (These assumptions were also made when introducing the problem in the main paper.)

$$d \geq 100, \qquad \frac{\alpha}{\sigma} \geq 10. \tag{39}$$

We define $P_p$ as the probability that for any given input $x \sim \mathcal{N}(y\boldsymbol{\mu}, \boldsymbol{\Sigma})$, the classifier $M_\mathcal{A}$ outputs the correct label y for the input $x + \delta_p$.

**p $= \infty$**

$$
\begin{aligned}
P_{\infty,\infty} &= \mathbb{P}_{x \sim \mathcal{N}(y\boldsymbol{\mu}, \boldsymbol{\Sigma})}[y \cdot M_{\ell_\infty}(x + \delta_\infty) > 0], \\
&= \mathbb{P}_{x \sim \mathcal{N}(y\boldsymbol{\mu}, \boldsymbol{\Sigma})}[y \cdot (x + \delta_\infty)^\top \mathbf{W} > 0], \\
&= \mathbb{P}_{x \sim \mathcal{N}(y\boldsymbol{\mu}, \boldsymbol{\Sigma})}[y \cdot (x_0 + \delta_\infty(0)) > 0], \\
&\geq \mathbb{P}_{x \sim \mathcal{N}(\boldsymbol{\mu}, \boldsymbol{\Sigma})}[x_0 > \epsilon_\infty]; \\
z &= \frac{\epsilon_\infty - \alpha}{\sigma} = \frac{\alpha}{\sigma}\left(\frac{1}{\sqrt{d}} - 1\right) + \frac{2}{\sqrt{d}}.
\end{aligned}
\tag{40}
$$

using the assumptions in (39),

$$P_{\infty,\infty} \geq 0.999. \tag{41}$$

**p = 1**

$$
\begin{aligned}
P_{1,1} &= \mathbb{P}_{x\sim\mathcal{N}(y\boldsymbol{\mu},\boldsymbol{\Sigma})}[y \cdot M_{\ell_1}(x+\delta_1) > 0], \\
&= \mathbb{P}_{x\sim\mathcal{N}(y\boldsymbol{\mu},\boldsymbol{\Sigma})}[y \cdot (x+\delta_1)^\top \mathbf{W} > 0], \\
&= \mathbb{P}_{x\sim\mathcal{N}(y\boldsymbol{\mu},\boldsymbol{\Sigma})}\left[y \cdot \frac{1}{\sqrt{d}}\sum_{i=1}^{d}(x_i+\delta_1(i)) > 0\right], \\
&= \mathbb{P}_{x\sim\mathcal{N}(y\boldsymbol{\mu},\boldsymbol{\Sigma})}[y \cdot (x_M+\delta_M) > 0], \\
&\geq \mathbb{P}_{x\sim\mathcal{N}(\boldsymbol{\mu},\boldsymbol{\Sigma})}\left[x_M > \frac{\epsilon_1}{\sqrt{d}}\right]; \\
z &= \frac{\frac{\epsilon_1}{\sqrt{d}}-\alpha}{\sigma} = \frac{\alpha}{\sigma}\left(\frac{1}{\sqrt{d}}-1\right) + \frac{2}{\sqrt{d}}.
\end{aligned}
\tag{42}
$$

using the assumptions in (39),

$$
P_{1,1} \geq 0.999.
\tag{43}
$$

## C.4 DECISION RULE FOR $C_{adv}$

We aim to provide a lower bound on the worst-case accuracy of the entire pipeline, through the existence of a simple decision tree $C_{adv}$. For given perturbation budgets $\epsilon_1$ and $\epsilon_\infty$, we aim to understand the range of values that can be taken by the adversarial input. Consider the scenarios described in Table 4 below. The same is also corroborated via the empirical experiments shown in Figure 4.

Table 4: The table shows the range of the values that the mean can take depending on the decision taken by the adversary. $\mu_0^{adv}$ and $\mu_M^{adv}$ represent the new mean of the distribution of features $x_0$ and $x_M$ after the adversarial perturbation.

| Attack Type | $\mu_0^{adv}$ | | $\mu_M^{adv}$ | |
|---|---|---|---|---|
| | y = 1 | y = -1 | y = 1 | y = -1 |
| None | $\alpha$ | $-\alpha$ | $\eta\sqrt{d}$ | $-\eta\sqrt{d}$ |
| $\ell_\infty$ | $\{\alpha-\epsilon_\infty, \alpha+\epsilon_\infty\}$ | $\{-\alpha-\epsilon_\infty, -\alpha+\epsilon_\infty\}$ | $\{\eta\sqrt{d}+\epsilon_\infty\sqrt{d}, \eta\sqrt{d}-\epsilon_\infty\sqrt{d}\}$ | $\{-\eta\sqrt{d}+\epsilon_\infty\sqrt{d}, -\eta\sqrt{d}-\epsilon_\infty\sqrt{d}\}$ |
| $\ell_1$ | $\{\alpha-\epsilon_1, \alpha+\epsilon_1\}$ | $\{-\alpha-\epsilon_1, -\alpha+\epsilon_1\}$ | $\{\eta\sqrt{d}+\epsilon_1/\sqrt{d}, \eta\sqrt{d}-\epsilon_1/\sqrt{d}\}$ | $\{-\eta\sqrt{d}+\epsilon_1/\sqrt{d}, -\eta\sqrt{d}-\epsilon_1/\sqrt{d}\}$ |

Note that any adversary that moves the perturbation away from the y-axis is uninteresting for our comparison, since irrespective of a correct perturbation type prediction by $C_{adv}$, either of the two second level models naturally obtain a high accuracy on such inputs. Hence, we define the following decision rule with all the remaining cases mapped to $\ell_1$ perturbation type.

$$
C_{adv}(x) = \begin{cases} 1, & \text{if} \quad ||x_0|-\alpha| < \epsilon_\infty + \frac{\alpha}{2} \\ 0, & \text{otherwise} \end{cases}
\tag{44}
$$

where the output 1 corresponds to the classifier predicting the presence of $\ell_\infty$ perturbation in the input, while an output of 0 suggests that the classifier predicts the input to contain perturbations of the $\ell_1$ type.

If we consider a black-box setting where the adversary has no knowledge of the classifier $C_{adv}$, and can only attack $M_\mathcal{A}$ it is easy to see that the proposed pipeline obtains a high adversarial accuracy against the union of $\ell_1$ and $\ell_\infty$ perturbations (since the given decision rule correctly classifies known examples as simulated in Figure 4.

Note: (1) There exists a single model that can also achieve robustness against the union of $\ell_1$ and $\ell_\infty$ perturbations, however, learning this model may be more challenging in real data settings. (2) The classifier need not be perfect.

## C.5 TRADE-OFF BETWEEN ATTACKING $M_\mathcal{A}$ AND $C_{adv}$

To obtain true robustness it is important that the entire pipeline is robust against adversarial attacks. More specifically, in this section we demonstrate the natural tension that exists between fooling the top level attack classifier (by making an

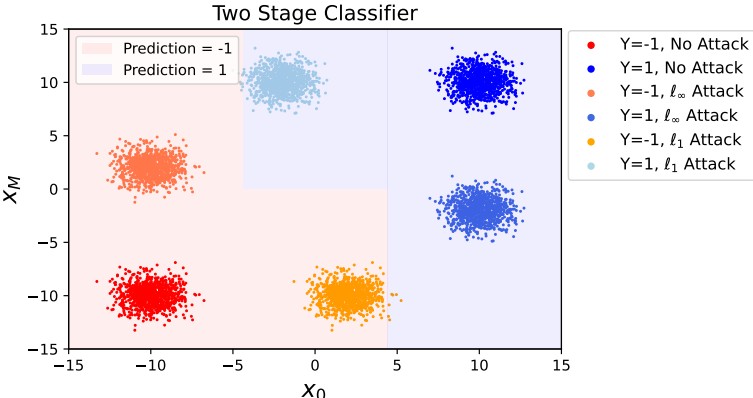

Figure 5: **Simulation**: Decision boundary of the overall two stage classifier. $x_M$ represents the meta feature as defined in Equation 37 and $x_0$ is the first dimension of the input.

adversarial attack less representative of its natural distribution) and fooling the bottom level adversarially robust models (requiring stronger attacks leading to a return to the attack's natural distribution).

The accuracy of the pipelined model $f$ against any input-label pair $(x, y)$ sampled through some distribution $\mathcal{N}(y\boldsymbol{\mu}_{adv}, \boldsymbol{\Sigma})$ (where $\boldsymbol{\mu}_{adv}$ incorporates the change in the input distribution owing to the adversarial perturbation) is given by:

$$
\begin{aligned}
\mathbb{P}\left[f(x) = y\right] &= \mathbb{P}_{x \sim \mathcal{N}(y\boldsymbol{\mu}_{adv}, \boldsymbol{\Sigma})}\left[C_{adv}(x)\right]\mathbb{P}_{x \sim \mathcal{N}(y\boldsymbol{\mu}_{adv}, \boldsymbol{\Sigma})}\left[y \cdot M_{\ell_\infty}(x) > 0 | C_{adv}(x)\right] \\
&\quad + (1 - \mathbb{P}_{x \sim \mathcal{N}(y\boldsymbol{\mu}_{adv}, \boldsymbol{\Sigma})}\left[C_{adv}(x)\right])\mathbb{P}_{x \sim \mathcal{N}(y\boldsymbol{\mu}_{adv}, \boldsymbol{\Sigma})}\left[y \cdot M_{\ell_1}(x) > 0 | \neg C_{adv}(x)\right], \\
&= \mathbb{P}_{x \sim \mathcal{N}(\boldsymbol{\mu}_{adv}, \boldsymbol{\Sigma})}\left[C_{adv}(x)\right]\mathbb{P}_{x \sim \mathcal{N}(\boldsymbol{\mu}_{adv}, \boldsymbol{\Sigma})}\left[M_{\ell_\infty}(x) > 0 | C_{adv}(x)\right] \\
&\quad + (1 - \mathbb{P}_{x \sim \mathcal{N}(\boldsymbol{\mu}_{adv}, \boldsymbol{\Sigma})}\left[C_{adv}(x)\right])\mathbb{P}_{x \sim \mathcal{N}(\boldsymbol{\mu}_{adv}, \boldsymbol{\Sigma})}\left[M_{\ell_1}(x) > 0 | \neg C_{adv}(x)\right].
\end{aligned}
\tag{45}
$$

$\ell_\infty$ **adversary.** To simplify the analysis, we consider loose lower bounds on the accuracy of the model $f$ against the $\ell_\infty$ adversary. Recall that the decision of the attack classifier is only dependent of the input $x_0$. Irrespective of the input features $x_i \forall i > 0$, it is always beneficial for the adversary to perturb the input by $\mu_i = -\epsilon_\infty$. However, the same does not apply for the input $x_0$. Analyzing for the scenario when the true label $y = 1$, if the input $x_0$ lies between $\frac{\alpha}{2} + \epsilon_\infty$ of the mean $\alpha$, irrespective of the perturbation, the output of the attack classifier $C_{adv} = 1$. The $M_{\ell_\infty}$ model then always correctly classifies these inputs. The overall robustness of the pipeline requires analysis for the case when input lies outside $\frac{\alpha}{2} + \epsilon_\infty$ of the mean as well. However, we consider that the adversary always succeeds in such a case in order to only obtain a loose lower bound on the robust accuracy of the pipeline model $f$ against $\ell_\infty$ attacks.

$$
\begin{aligned}
\mathbb{P}\left[f(x) = y\right] &= \mathbb{P}_{x \sim \mathcal{N}(\boldsymbol{\mu}_{adv}, \boldsymbol{\Sigma})}\left[C_{adv}(x)\right]\mathbb{P}_{x \sim \mathcal{N}(\boldsymbol{\mu}_{adv}, \boldsymbol{\Sigma})}\left[M_{\ell_\infty}(x) > 0 | C_{adv}(x)\right], \\
&\quad + (1 - \mathbb{P}_{x \sim \mathcal{N}(\boldsymbol{\mu}_{adv}, \boldsymbol{\Sigma})}\left[C_{adv}(x)\right])\mathbb{P}_{x \sim \mathcal{N}(\boldsymbol{\mu}_{adv}, \boldsymbol{\Sigma})}\left[M_{\ell_1}(x) > 0 | \neg C_{adv}(x)\right], \\
&\geq \mathbb{P}_{x \sim \mathcal{N}(\boldsymbol{\mu}_{adv}, \boldsymbol{\Sigma})}\left[C_{adv}(x)\right]\mathbb{P}_{x \sim \mathcal{N}(\boldsymbol{\mu}_{adv}, \boldsymbol{\Sigma})}\left[M_{\ell_\infty}(x) > 0 | C_{adv}(x)\right], \\
&\geq \mathbb{P}_{x \sim \mathcal{N}(\boldsymbol{\mu}, \boldsymbol{\Sigma})}\left[|x_0 - \alpha| \leq \frac{\alpha}{2} - \epsilon_\infty\right], \\
&\geq 2\mathbb{P}_{x \sim \mathcal{N}(\boldsymbol{\mu}, \boldsymbol{\Sigma})}\left[x_0 \leq \alpha - \frac{\alpha}{2} + \epsilon_\infty\right], \\
z &= \frac{(\alpha - \frac{\alpha}{2} + \epsilon_\infty) - \alpha}{\sigma} = -\frac{\alpha}{2\sigma} + \frac{3\sigma}{2\sigma\sqrt{d}}.
\end{aligned}
\tag{46}
$$

using the assumptions in (39),

$$
\mathbb{P}\left[f(x) = y\right] \sim 0.99.
\tag{47}
$$

$\ell_1$ **adversary.** It may be noted that a trivial way for the $\ell_1$ adversary to fool the attack classifier is to return a perturbation $\delta_1 = 0$. In such a scenario, the classifier predicts that the adversarial image was subjected to an $\ell_\infty$ attack. The label prediction is hence made by the $M_{\ell_\infty}$ model. But we know from (41) that the $M_{\ell_\infty}$ model predicts benign inputs correctly with a probability $P_{\infty,\infty} > 0.99$, hence defeating the adversarial objective of misclassification. To achieve misclassification

over the entire pipeline the optimal perturbation decision for the $\ell_1$ adversary when $x_0 \in \left[ -\alpha - \frac{\alpha}{2} - \epsilon_1, -\alpha + \frac{\alpha}{2} + \epsilon_1 \right]$ the adversary can fool the pipeline by ensuring that the $C_{adv}(x) = 1$. However, in all the other cases irrespective of the perturbation, either $C_{adv} = 0$ or the input features $x_0$ has the same sign as the label $y$. Since, $P_{1,1} > 0.99$ for the $M_{\ell_1}$ model, for all the remaining inputs $x_0$ the model correctly predicts the label with probability greater than 0.99 (approximate lower bound). We formulate this trade-off to elaborate upon the robustness of the proposed pipeline.

$$
\begin{aligned}
\mathbb{P}\left[f(x) = y\right] &= \mathbb{P}_{x \sim \mathcal{N}(\boldsymbol{\mu}_{adv}, \boldsymbol{\Sigma})}\left[C_{adv}(x)\right] \mathbb{P}_{x \sim \mathcal{N}(\boldsymbol{\mu}_{adv}, \boldsymbol{\Sigma})}\left[M_{\ell_\infty}(x) > 0 | C_{adv}(x)\right] \\
&\quad + (1 - \mathbb{P}_{x \sim \mathcal{N}(\boldsymbol{\mu}_{adv}, \boldsymbol{\Sigma})}\left[C_{adv}(x)\right]) \mathbb{P}_{x \sim \mathcal{N}(\boldsymbol{\mu}_{adv}, \boldsymbol{\Sigma})}\left[M_{\ell_1}(x) > 0 | \neg C_{adv}(x)\right], \\
&\geq \mathbb{P}_{x \sim \mathcal{N}(\boldsymbol{\mu}, \boldsymbol{\Sigma})}\left[-\alpha - \frac{\alpha}{2} - \epsilon_1 \leq x_0 \leq -\alpha + \frac{\alpha}{2} + \epsilon_1\right] \\
&\quad + 0.999(\mathbb{P}_{x \sim \mathcal{N}(\boldsymbol{\mu}, \boldsymbol{\Sigma})}\left[x_0 < -\alpha - \frac{\alpha}{2} - \epsilon_1 \text{ or } x_0 > -\alpha + \frac{\alpha}{2} + \epsilon_1\right]), \\
&\geq 0.999(\mathbb{P}_{x \sim \mathcal{N}(\boldsymbol{\mu}, \boldsymbol{\Sigma})}\left[x_0 < -\alpha - \frac{\alpha}{2} - \epsilon_1 \text{ or } x_0 > -\alpha + \frac{\alpha}{2} + \epsilon_1\right]).
\end{aligned}
\tag{48}
$$

using the assumptions in (39),

$$
\mathbb{P}\left[f(x) = y\right] \sim 0.99. \tag{49}
$$

This concludes the proof for Theorem 2, showing that an adversary can hardly stage successful attacks on the entire pipeline and faces a natural tension between attacking the label predictor and the attack classifier. We verify these results via a simulation in Figure 5. We emphasize that these accuracies are lower bounds on the actual robust accuracy, and the objective of this analysis is not to find the optimal solution to the problem of multiple perturbation adversarial training, but to elucidate the trade-off between attacking the two pipeline stages.

## D  MODEL ARCHITECTURE

**Second-level $M_\mathcal{A}$ models.**    A key advantage of PROTECTOR is that we can build upon existing defenses against individual perturbation type. Specifically, for MNIST, we use the same CNN architecture as Zhang et al. [2019] for our $M_\mathcal{A}$ models, and we train these models using their proposed TRADES loss. For CIFAR-10, we use the same training setup and model architecture as Carmon et al. [2019], which is based on a robust self-training algorithm that utilizes unlabeled data to improve the model robustness.

**Perturbation classifier $C_{adv}$.**    For both MNIST and CIFAR-10 datasets, the architecture of the perturbation classifier $C_{adv}$ is similar to the individual $M_\mathcal{A}$ models. Specifically, for MNIST, we use the CNN architecture in Zhang et al. [2019] with four convolutional layers, followed by two fully-connected layers. For CIFAR-10, $C_{adv}$ is a WideResNet [Zagoruyko and Komodakis, 2016] model with depth 16 and widening factor of 2 (WRN-16-2). The architectures for classifying $\ell_p$ perturbations and common corruptions are largely the same, except that the final classification layers have different dimensions due to the different label set sizes.

## E  TRAINING DETAILS

### E.1  SPECIALIZED ROBUST PREDICTORS $M_\mathcal{A}$

**MNIST.**    We use the Adam optimizer [Kingma and Ba, 2015] to train our models along with a piece-wise linearly varying learning rate schedule [Smith, 2018] to train our models with maximum learning rate of $10^{-3}$. The base models $M_{\ell_1}, M_{\ell_2}, M_{\ell_\infty}$ are trained using the TRADES algorithm for 20 iterations, and step sizes $\alpha_1 = 2.0$, $\alpha_2 = 0.3$, and $\alpha_\infty = 0.05$ for the $\ell_1, \ell_2, \ell_\infty$ attack types within perturbation radii $\epsilon_1 = 10.0$, $\epsilon_2 = 2.0$, and $\epsilon_\infty = 0.3$ respectively.[4]

**CIFAR10.**    The individual $M_\mathcal{A}$ models are trained to be robust against $\{\ell_\infty, \ell_1, \ell_2\}$ perturbations of $\{\epsilon_\infty, \epsilon_1, \epsilon_2\} = \{0.003, 10.0, 0.05\}$ respectively. For CIFAR10, the attack step sizes $\{\alpha_\infty, \alpha_1, \alpha_2\} = \{0.005, 2.0, 0.1\}$ respectively. The training of the individual $M_\mathcal{A}$ models is directly based on the work of Carmon et al. [2019].

---

[4]We use the Sparse $\ell_1$ descent [Tramèr and Boneh, 2019] for the PGD attack in the $\ell_1$ constraint.

### E.2    PERTURBATION CLASSIFIER $C_{adv}$

**MNIST.**    We train the model for 5 epochs using the SGD optimizer with weight decay as $5 \times 10^{-4}$. We used a variation of the learning rate schedule from Smith [2018], which is piecewise linear from $5 \times 10^{-4}$ to $10^{-3}$ over the first 2 epochs, and down to 0 till the end. The batch size is set to 100 for all experiments.

**CIFAR10.**    We train the model for 5 epochs using the SGD optimizer with weight decay as $5 \times 10^{-4}$. We used a variation of the learning rate schedule from Smith [2018], which is piecewise linear from $5 \times 10^{-3}$ to $10^{-2}$ over the first 2 epochs, and down to 0 till the end. The batch size is set to 100 for all experiments.

**Creating the Adversarial Perturbation Dataset.**    We create a static dataset of adversarially perturbed images and their corresponding attack label for training the perturbation classifier $C_{adv}$. For generating adversarial images, we perform weak adversarial attacks that are faster to compute. In particular, we perform 10 iterations of the PGD attack. For MNIST, the attack step sizes $\{\alpha_\infty, \alpha_1, \alpha_2\} = \{0.05, 2.0, 0.3\}$ respectively. For CIFAR10, the attack step sizes $\{\alpha_\infty, \alpha_1, \alpha_2\} = \{0.005, 2.0, 0.1\}$ respectively. Note that we perform the Sparse-$\ell_1$ or the top-k PGD attack for the $\ell_1$ perturbation ball, as introduced by Tramèr and Boneh [2019]. We set the value of k to 10, that is we move by a step size $\frac{\alpha_1}{k}$ in each of the top 10 directions with respect to the magnitude of the gradient.

**CIFAR10-C.**    We use a dropout value of 0.3 along with the same optimizer (SGD). We use a learning rate of 0.01 and SGD optimizer for 5 epochs, with linear rate decay to 0.001 between the second epoch and the fifth epoch For experiments on classifying corruptions of severity 1, we find that the model takes longer to train. Hence, we train the model for 10 epochs, whereas all other models (at other severity levels) were trained for 5 epochs.

## F    ATTACKS USED FOR EVALUATION

A description of all the attacks used for evaluation of the models is presented here. From the AutoAttack library [Croce and Hein, 2020b], we make use of all the three variants of the Adaptive PGD attack (APGD-CE, APGD-DLR, APGD-T) along with the targeted and standard version of Fast Adaptive Boundary Attack (FAB, FAB-T) [Croce and Hein, 2020a] and the Square Attack [Andriushchenko et al., 2020]. We utilize the AA$^+$ version in the auto-attack library for stronger attacks.

**Attack Hyperparameters.**    For the attacks in the AutoAtack library we use the default parameter setting in the strongest available mode (such as AA$^+$). For the custom PGD attacks, we evaluate the models with 10 restarts and 200 iterations of the PGD attack. The step size of the $\{\ell_\infty, \ell_1, \ell_2\}$ PGD attacks are set as follows: For MNIST, the attack step sizes $\{\alpha_\infty, \alpha_1, \alpha_2\} = \{0.01, 1.0, 0.1\}$ respectively. For CIFAR10, the attack step sizes $\{\alpha_\infty, \alpha_1, \alpha_2\} = \{0.003, 1.0, 0.02\}$ respectively.

Further, in line with previous work [Tramèr and Boneh, 2019, Maini et al., 2020] we evaluate our models on the first 1000 images of the test set of MNIST and CIFAR-10, since many of the attacks employed are extremely computationally expensive and slow to run. Specifically, on a single GPU, the entire evaluation for a single model against all the attacks discussed with multiple restarts will take nearly 1 month, and is not feasible.

## G    FOURIER FEATURES

Yin et al. [2019a] studied various perturbations in their Fourier domain. Their work mainly focused on studying the Fourier spectrum of various common corruptions, and they showed how model robustness was affected by the data augmentation scheme used. In particular, they found that certain augmentation strategies benefit robustness to perturbations in the high frequency domain.

On the contrary, in our work, we use Fourier features to classify perturbation types. While Yin et al. [2019a] directly studied only the perturbation ($\delta$) added to the image, we visualize the Fourier transform of the actual perturbed image ($\mathbf{x} + \delta$). This makes it more challenging to distill the perturbation from the original image. Secondly, we study the Fourier transform of various adversarially crafted examples. In what follows, we will first provide a visual example to justify how adversarial examples crafted by different attack types, have different Fourier spectrums. We then utilise this property to use Fourier features as an input to the perturbation classifier for classifying the perturbation type.

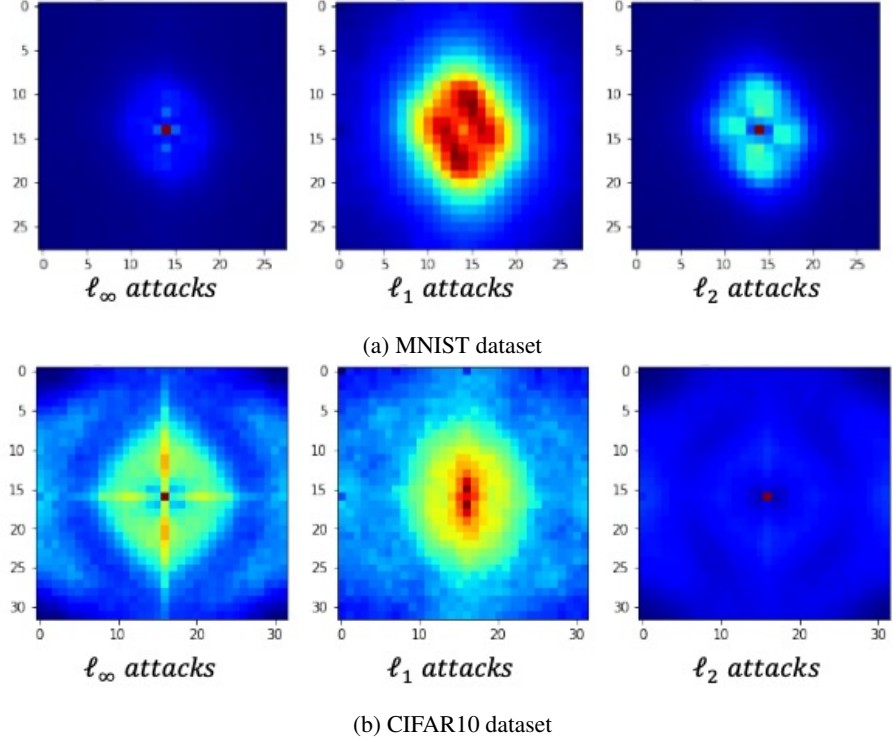

(a) MNIST dataset

(b) CIFAR10 dataset

Figure 6: We present the Fourier spectrums of various attacks on a vanilla model trained on (a) MNIST and (b) CIFAR10 datasets by averaging the per-pixel DFT over the entire test set, i.e. for an $\ell_\infty, \ell_1, \ell_2$ adversarial example corresponding to image in the test set.

**Fourier Spectrum.** We follow the same naming convention as Yin et al. [2019a]. For an input image $\mathbf{x} \in \mathbb{R}^{d_1 \times d_2}$, we will represent the 2-dimensional discrete Fourier transform (DFT) by $\mathcal{F} : \mathbb{R}^{d_1 \times d_2} \to \mathbb{C}^{d_1 \times d_2}$. $\mathcal{F}^{-1}$ represents the inverse DFT. Since the Fourier transform belongs to the complex plane, we estimate $\mathbb{E}\left[|\mathcal{F}(\mathbf{x}_{adv})[i,j]|\right]$ by averaging over adversarial examples generate for each image in the test set.

Note that Yin et al. [2019a] had estimated only the perturbation ($\mathbb{E}\left[|\mathcal{F}(\mathbf{x}_{adv} - \mathbf{x})[i,j]|\right]$) and not the perturbed image in their work. However, since at test time we do not have access to the original image, we only perform our analysis based on the perturbed input.

We present the Fourier spectrums in Figure 6. While adversarial examples typically have an imperceptible amount of perturbation for the human eye, the visualization of these adversarial examples through the Fourier spectrums help us visually distinguish between them. We also note that the Fourier spectrum for each attack does not show similar characteristics across different datasets (MNIST and CIFAR10). However, the characteristics stay consistent when independently attacking a given model on the same dataset.

We use this observation to augment PROTECTOR with an ensemble of diverse perturbation classifiers. We do so by training another model $C_{adv}$ for which the inputs are *only* the Fourier features of the corresponding adversarial examples. The training process and architecture for such a classifier stays identical as one that classifies adversarial examples in their image domain.

# H    PERTURBATION CATEGORIZATION

## H.1    EMPIRICAL PERTURBATION OVERLAP

While we justify the choice of perturbation sizes in our theoretical proofs in Appendix B.4 and C.2, in this section we demonstrate the empirical agreement of the choices of perturbation sizes we make for our results on MNIST and CIFAR10 datasets. To measure how often adversarial perturbations of different attacks overlap, we empirically quantify the overlapping regions by attacking a benign model with PGD attacks. In Table 5 we report the range of the norm of perturbations in the

Table 5: **Vanilla Model:** Empirical overlap of $\ell_{p,\epsilon_p}$ attack perturbations in different $\ell_{q,\epsilon_q}$ regions for (a) MNIST $(\epsilon_1, \epsilon_2, \epsilon_\infty) = (10, 2.0, 0.3)$; (b) CIFAR-10 $(\epsilon_1, \epsilon_2, \epsilon_\infty) = (10, 0.5, 0.03)$. Each column represents the range (min - max) of $\ell_q$ norm for perturbations generated using $\ell_p$ PGD attack.

| Attack | MNIST | | | CIFAR10 | | |
|---|---|---|---|---|---|---|
| | $\ell_\infty < 0.3$ | $\ell_2 < 2.0$ | $\ell_1 < 10$ | $\ell_\infty < 0.03$ | $\ell_2 < 0.5$ | $\ell_1 < 10$ |
| PGD $\ell_\infty$ | $\leq 0.3$ | (3.67 - 6.05) | (54.8 - 140.9) | $\leq 0.03$ | (1.33 - 1.59) | (62.7 - 85.5) |
| PGD $\ell_2$ | (0.40 - 0.86) | $\leq 2.0$ | (11.2 - 24.1) | (0.037 - 0.10) | $\leq 0.05$ | (15.4 - 20.9) |
| Sparse $\ell_1$ | (0.70 - 1.0) | (2.08 - 2.92) | $\leq 10.0$ | (0.27 - 0.77) | (1.32 - 1.88) | $\leq 10.0$ |

Table 6: **PROTECTOR:** Empirical overlap of $\ell_{p,\epsilon_p}$ attack perturbations in different $\ell_{q,\epsilon_q}$ regions for (a) MNIST $(\epsilon_1, \epsilon_2, \epsilon_\infty) = (10, 2.0, 0.3)$; (b) CIFAR-10 $(\epsilon_1, \epsilon_2, \epsilon_\infty) = (10, 0.5, 0.03)$. Each column represents the range (min - max) of $\ell_q$ norm for perturbations generated using $\ell_p$ PGD attack.

| Attack | MNIST | | | CIFAR10 | | |
|---|---|---|---|---|---|---|
| | $\ell_\infty < 0.3$ | $\ell_2 < 2.0$ | $\ell_1 < 10$ | $\ell_\infty < 0.03$ | $\ell_2 < 0.5$ | $\ell_1 < 10$ |
| PGD $\ell_\infty$ | $\leq 0.3$ | (5.03-6.12) | (100.40-138.52) | $\leq 0.03$ | (1.46-1.69) | (73.15-93.26) |
| PGD $\ell_2$ | (0.35-0.95) | $\leq 2.0$ | (17.06-27.88) | (0.036-0.29) | $\leq 0.05$ | (5.83-21.21) |
| Sparse $\ell_1$ | (0.81-1.0) | (2.13-2.98) | $\leq 10.0$ | (0.42-1.0) | (1.50-2.91) | $\leq 10.0$ |

alternate perturbation region for any given attack type. The observed overlap is exactly 0% in all cases and the observation is consistent across MNIST and CIFAR10 datasets.

Table 7: Perturbation type classification accuracy for different perturbation types. The perturbation classifier $C_{adv}$ is trained on adversarial examples against two $M_A$ models. Each column represent the model used to create transfer-based attack via the attack type in the corresponding row. The represented accuracy is an aggregate over 1000 randomly sampled attacks of the $\ell_\infty, \ell_2, \ell_1$ types for the corresponding algorithms (and datasets).

| | $M_{\ell_\infty}$ | $M_{\ell_2}$ | $M_{\ell_1}$ | MAX | AVG | MSD |
|---|---|---|---|---|---|---|
| MNIST-PGD | 100% | 100% | 99.3% | 99.0% | 99.6% | 99.1% |
| MNIST-AutoAttack | 100% | 100% | 99.0% | 99.5% | 100% | 100% |
| CIFAR10-PGD | 99.9% | 99.5% | 100% | 100% | 98.7% | 95.7% |
| CIFAR10-AutoAttack | 99.9% | 99.9% | 100% | 100% | 99.7% | 99.7% |

To contrast the results with that of attacking a vanilla model, we also present results on the perturbation overlap when we attack PROTECTOR with PGD attacks (in Table 6). It is noteworthy that the presence of a perturbation classifier forces the adversaries to generate such attacks that increase the norm of the perturbations in alternate $\ell_q$ region. Secondly, we also observe that in the case of CIFAR10, the $\ell_2$ PGD attack has a large overlap with the $\ell_1$ norm of radius 10. However, recall that in case of $\ell_2$ attacks for CIFAR10, both the base models $M_{\ell_1}$ and $M_{\ell_\infty}$ were satisfactorily robust. Hence, the attacker has no incentive to reduce the perturbation radius for an $\ell_q$ norm since the perturbation classifier only performs a binary classification between $\ell_1$ and $\ell_\infty$ attacks.

## H.2 ROBUSTNESS OF $C_{adv}$

In this section, we present the results of the perturbation type classifier $C_{adv}$ against transfer adversaries. The results for the robustness of the perturbation classifier $C_{adv}$ in the presence of adaptive adversaries is presented in Table 7. Note that $C_{adv}$ transfers well across the board, even if the adversarial examples are generated against new models that are unseen for $C_{adv}$ during training, achieving extremely high test accuracy. Further, even if the adversarial attack was generated by a different algorithm such as from the AutoAttack library, the transfer success of $C_{adv}$ still holds up. In particular, the obtained accuracy is $> 95\%$ across all the individual test sets created. The attack classification accuracy is in general highest against those generated by attacking $M_{\ell_1}$ or $M_{\ell_\infty}$ for CIFAR10, and $M_{\ell_2}$ or $M_{\ell_\infty}$ for MNIST. This is an expected consequence of the nature of generation of the static dataset for training the perturbation classifier $C_{adv}$ as described in Section 5.1.

Table 8: Classification accuracy for common corruptions at different severity levels. The task is a 19 class classification problem. In the training setting "Combined", all images of different severity levels are used for training. The model predicts the corruption type among the 19 possible corruptions.

| Training | Tested on | | | | |
| --- | --- | --- | --- | --- | --- |
| | Level 1 | Level 2 | Level 3 | Level 4 | Level 5 |
| Level Specific | 87.2% | 97.7% | 97.0% | 98.7% | 99.5% |
| Combined | 85.4% | 96.2% | 97.2% | 98.1% | 99.1% |

Table 9: Comparison between using a 'softmax' based aggregation of predictions from different specialized models versus using the prediction from the model corresponding to the most likely attack (only at inference time). Results are presented for APGD $\ell_2, \ell_\infty$ attacks on the CIFAR10 dataset.

| Attack | Max-approach (Eq. 3) | Softmax-approach (Eq. 4) |
| --- | --- | --- |
| APGD-CE $\ell_2$ ($\epsilon_2 = 0.5$) | 75.7% | 75.6% |
| APGD-DLR $\ell_2$ ($\epsilon_2 = 0.5$) | 76.5% | 76.7% |
| APGD-CE $\ell_\infty$ ($\epsilon_\infty = 0.03$) | 86.9% | 86.9% |
| APGD-DLR $\ell_\infty$ ($\epsilon_\infty = 0.03$) | 91.8% | 91.2% |

## H.3 MORE RESULTS ON COMMON CORRUPTIONS

For each image in the original CIFAR-10 test set, CIFAR-10-C includes corrupted images of 19 different corruption types at 5 severity levels. In this section, we present results on corruption classification at different severity levels. Specifically, we train a single model on images of all severity levels. Then to evaluate on each of the 5 severity levels, we also train another model on corrupted images of the same level. As mentioned in Section 6.1, each corruption type has 9K training samples at each severity level, and 1K for testing. We ensure that all corrupted samples of the same original CIFAR-10 image are in the same data split, so that no sample in the test split corresponds to the same original image in the training split.

We present the corruption type classification accuracies at different severity levels in Table 8. We observe that the classification accuracy is around 90% for all severity levels, even when the severity level is low and the corruptions are hard to notice for the human eye. Note that for a 19-class classification problem, random guessing would only yield about 5% accuracy. Further, the test accuracy increases as the severity of the corruption increases. This can be explained due the fact that increasing the magnitude of corruptions makes them more representative and easier to be distinguished from others. Note that models trained on standard image classification tasks are typically more resilient to corruptions at a lower severity, and images with a high corruption severity can be detrimental to the prediction performance of standard classifiers. Therefore, it is important to correctly identify such highly corrupted images. We also note that a combined model trained on multiple corruption severity levels does not have a significant trade-off in test accuracy to those trained on the specific levels. Specifically, the drop in test set accuracy varies between 0.4% and 1.8% across various severity levels, and the decrease is much less noticeable when the severity level becomes large.

## I ADAPTIVE ATTACKS

### I.1 AGGREGATING PREDICTIONS FROM DIFFERENT $M_{\mathcal{A}}$ AT INFERENCE

In all our experiments in this work the adversary constructs adversarial examples using the softmax based adaptive strategy for aggregating predictions from different $M_{\mathcal{A}}$ models, as described in Equation 4 for the column 'Ours' and using the 'max' strategy (Equation 3) for results described in the column 'Ours*'

However, for consistency of our defense strategy irrespective of the attacker's strategy, the defender only utilizes predictions from the specialized model $M_{\mathcal{A}}$ corresponding to the most-likely attack (Equation 3) to provide the final prediction (only forward propagation) for generated adversarial examples. In our evaluation, we found a negligible impact of changing this aggregation to the 'softmax' strategy for aggregating the predictions. For example, we show representative results in case of the APGD ($\ell_\infty, \ell_2$) attacks on the CIFAR10 dataset in Table 9.

Table 10: Performance of Adaptive attacks that attempt to separately fool the perturbation classifier and the alternate specialized robust model. The corresponding objective functions for each attack are specified in Appendix I.

| Attack | Dual Attack (Eq. 51) | Binary Attack (Eq. 52) |
|---|---|---|
| PGD $\ell_\infty$ ($\epsilon_\infty = 0.03$) | 69.3% | 73.2% |
| PGD $\ell_2$ ($\epsilon_2 = 0.5$) | 72.1% | 74.8% |
| Sparse PGD $\ell_1$ ($\epsilon_1 = 10$) | 64.7% | 59.1% |

## I.2 TRADE-OFF BETWEEN FOOLING $M_\mathcal{A}$ AND $C_{adv}$

The adversary chooses the strongest attack over a set of adaptive attacks targeted at each $M_\mathcal{A}$. For any data point (x,y) each targeted attack optimises the following constraint:

$$\min_{\delta_p} \ell_p(x + \delta_p)$$
$$\text{s.t.} \quad M_\mathcal{A}(x + \delta_p) \neq y; \quad C_{adv}(x + \delta_p) = p \tag{50}$$

We perform the attack for each of the PGD attacks for $p \in \{1, 2, \infty\}$. To design the exact objective function for optimization of Equation 50, we take inspiration from a similar exploration by Carlini and Wagner [2017b].

**First**, we combine a dual loss function for individually fooling the $M_\mathcal{A}$ model and the perturbation classifier $C_{adv}$ by giving different importance to each of them using a parameter $\lambda$. More specifically, for an input $(x, y)$, the objective for finding an adversarial example of type $\mathcal{A} \in \mathcal{S}$ can be written as:

$$\mathcal{L}_{(x,y,\mathcal{A})} = -1 \cdot \text{CrossEntropyLoss}(C_{adv}(x), \mathcal{A}) + \lambda \cdot \text{CrossEntropyLoss}(\mathcal{M}_\mathcal{B}(x), y) \tag{51}$$

where $\mathcal{B} = \arg\max C_{adv}(x)$. We experiment with values of $\lambda \in \{10^{-1}, 1, 10, 100\}$ and report the worst adversarial example in each case.

**Secondly**, we design an alternate approach where the adversary is constrained to fool the perturbation classifier (owing to a strong binary misclassification loss). It then attempts to fool the alternate $M_\mathcal{A}$ model under this constraint. More specifically, if $\mathcal{B} = \arg\max C_{adv}(x)$, then

$$\mathcal{L}_{(x,y,\mathcal{A})} = -1 \cdot (\mathcal{A} = \mathcal{B}) + \lambda \cdot \text{CrossEntropyLoss}(\mathcal{M}_\mathcal{B}(x), y) \tag{52}$$

We perform the above optimization for the PGD attacks in the $\ell_\infty, \ell_1, \ell_2$ perturbation radius constraints. In case of the $\ell_1$ attack, we optimize using the stronger Sparse-$\ell_1$ attack [Tramèr and Boneh, 2019]. The adversarial robustness of PROTECTOR (on CIFAR10) to these attacks is reported in Table 10. We note that the formulation used in the main paper (Equation 4) that uses a 'softmax' bridge between the two levels of the pipeline performs better than the attacks outlined above. In particular, we observe that adversaries find it difficult to balance the two losses separately in order to satisfy the dual constraint.

## J  BREAKDOWN OF COMPLETE EVALUATION

Now we present a breakdown of results of the adversarial robustness of baseline approaches and PROTECTOR against all the attacks in our suite. We also report the worst case performance against the union of all attacks.

### J.1  MNIST

In Table 11, we provide a breakdown of the adversarial accuracy of all the baselines, individual $M_\mathcal{A}$ models and the PROTECTOR method, with both the adaptive and standard attack variants on the MNIST dataset. PROTECTOR outperforms prior baselines by $6.4\%$ on the MNIST dataset. It is important to note that PROTECTOR shows significant improvements against most attacks in the suite. Compared to the previous state-of-the-art defense against multiple perturbation types

Table 11: Attack-wise breakdown of adversarial robustness on the MNIST dataset. *Ours* represents the PROTECTOR method against the adaptive attack strategy described in Section 5.4, and *Ours\** represents the standard attack setting.

| | $M_{\ell_\infty}$ | $M_{\ell_2}$ | $M_{\ell_1}$ | MAX | AVG | MSD | Ours | Ours* |
|---|---|---|---|---|---|---|---|---|
| Benign Accuracy | 99.2% | 98.7% | 98.8% | 98.6% | 99.1% | 98.3% | 98.9% | 98.9% |
| PGD-$\ell_\infty$ | 92.8% | 6.2% | 0.0% | 50.0% | 64.8% | 65.7% | 83.5% | 89.1% |
| APGD-CE | 91.5% | 3.6% | 0.0% | 41.0% | 59.1% | 65.2% | 84.3% | 84.6% |
| APGD-DLR | 91.8% | 8.0% | 0.0% | 43.9% | 61.9% | 66.0% | 88.6% | 88.4% |
| APGD-T | 91.9% | 2.9% | 0.0% | 39.6% | 59.0% | 64.4% | 88.0% | 88.6% |
| FAB-T | 92.5% | 5.0% | 0.0% | 48.8% | 64.3% | 65.5% | 99.0% | 98.6% |
| SQUARE | 90.3% | 7.6% | 0.0% | 45.9% | 65.1% | 68.2% | 93.0% | 93.3% |
| $\ell_\infty$ attacks ($\epsilon = 0.3$) | 90.2% | 2.6% | 0.0% | 39.0% | 57.8% | 63.5% | 78.1% | 79.0% |
| PGD-$\ell_2$ | 84.9% | 74.9% | 51.6% | 63.6% | 69.5% | 71.7% | 73.0% | 75.5% |
| DDN | 42.3% | 76.0% | 53.1% | 62.2% | 64.6% | 70.1% | 87.5% | 94.3% |
| APGD-CE | 78.9% | 74.0% | 50.7% | 61.9% | 65.0% | 69.6% | 72.2% | 76.4% |
| APGD-DLR | 79.3% | 75.2% | 54.1% | 63.2% | 65.1% | 70.9% | 74.4% | 78.2% |
| APGD-T | 80.7% | 73.8% | 48.0% | 61.0% | 63.9% | 69.6% | 70.8% | 74.3% |
| FAB-T | 12.2% | 74.8% | 49.4% | 62.5% | 63.7% | 69.1% | 86.9% | 96.3% |
| SQUARE | 25.6% | 82.3% | 66.6% | 71.7% | 71.8% | 75.0% | 96.9% | 96.6% |
| $\ell_2$ attacks ($\epsilon = 2.0$) | 9.5% | 72.3% | 47.8% | 58.5% | 58.6% | 65.7% | 66.6% | 72.3% |
| PGD-$\ell_1$ | 72.5% | 74.6% | 78.5% | 52.9% | 59.3% | 67.9% | 73.8% | 79.4% |
| FAB-T | 20.0% | 71.6% | 77.6% | 43.9% | 51.2% | 67.5% | 74.3% | 85.0% |
| $\ell_1$ attacks ($\epsilon = 10$) | 18.8% | 70.6% | 77.5% | 41.8% | 46.1% | 64.3% | 68.1% | 72.5% |
| All Attacks | 7.3% | 2.6% | 0.0% | 29.1% | 37.1% | 57.2% | 63.6% | 67.2% |
| Average All Attacks | 69.8% | 47.4% | 35.3% | 54.1% | 63.2% | 68.4% | 83.1% | 86.6% |

(MSD), if we compare the performance gain on each individual attack algorithm, the average accuracy increase of 14.7% on MNIST dataset. These results demonstrate that PROTECTOR considerably mitigates the trade-off in accuracy against individual attack types.

## J.2 CIFAR-10

In Table 12, we provide a breakdown of the adversarial accuracy of all the baselines, individual $M_A$ models and the PROTECTOR method, with both the adaptive and standard attack variants on the CIFAR10 dataset. PROTECTOR outperforms prior baselines by 10%. Once again, note that PROTECTOR shows significant improvements against most attacks in the suite. Compared to the previous state-of-the-art defense against multiple perturbation types (MSD), if we compare the performance gain on each individual attack algorithm, the improvement is significant, with an average accuracy increase of 14.2% on. These results demonstrate that PROTECTOR considerably mitigates the trade-off in accuracy against individual attack types. Further, PROTECTOR also retains a higher accuracy on benign images, as opposed to past defenses that have to sacrifice the benign accuracy for the robustness on multiple perturbation types. The clean accuracy of PROTECTOR is over 7% higher than such existing defenses on CIFAR-10, and the accuracy is close to $M_A$ models trained for a single perturbation type.

## J.3 DIFFERENT NUMBER OF SECOND-LEVEL $M_A$ PREDICTORS

We also evaluate PROTECTOR with three second-level predictors, i.e., $M_{\ell_1}$, $M_{\ell_2}$ and $M_{\ell_\infty}$. The results are presented in Table 13. This alternative design reduces the overall accuracy of the pipeline model. We hypothesize that this happens because the $M_{\ell_1}$ model is already reasonably robust against the $\ell_2$ attacks, as shown in Table 2b. However, having both $M_{\ell_1}$ and $M_{\ell_2}$ models allows adaptive adversaries to find larger regions for fooling both $C_{adv}$ and $M_A$, thus hurting the overall performance against adaptive adversaries.

Table 12: Attack-wise breakdown of adversarial robustness on CIFAR-10. *Ours* represents PROTECTOR against the adaptive attack strategy described in Section 5.4, and *Ours\** represents the standard attack setting.

| | $M_{\ell_\infty}$ | $M_{\ell_2}$ | $M_{\ell_1}$ | MAX | AVG | MSD | Ours | Ours* |
|---|---|---|---|---|---|---|---|---|
| Benign Accuracy | 89.5% | 93.9% | 89.0% | 81.0% | 84.6% | 81.7% | 89.0% | 89.0% |
| PGD-$\ell_\infty$ | 62.3% | 36.2% | 36.0% | 43.2% | 41.1% | 46.6% | 62.3% | 62.3% |
| APGD-CE | 62.1% | 35.5% | 35.9% | 38.5% | 41.1% | 46.3% | 62.2% | 63.9% |
| APGD-DLR | 60.9% | 38.0% | 37.7% | 39.1% | 43.3% | 46.6% | 59.1% | 63.8% |
| APGD-T | 59.4% | 34.9% | 35.0% | 36.5% | 39.7% | 43.8% | 58.7% | 62.3% |
| FAB-T | 59.9% | 35.9% | 35.4% | 40.8% | 40.2% | 44.0% | 79.1% | 84.7% |
| SQUARE | 67.2% | 57.7% | 50.5% | 51.8% | 50.8% | 52.1% | 85.6% | 80.3% |
| $\ell_\infty$ attacks ($\epsilon = 0.003$) | 59.3% | 34.8% | 35.0% | 34.9% | 39.7% | 43.7% | 56.1% | 58.4% |
| PGD-$\ell_2$ | 66.5% | 77.5% | 72.4% | 64.4% | 67.7% | 66.2% | 69.4% | 69.6% |
| DDN | 66.9% | 77.5% | 72.6% | 64.5% | 67.7% | 66.2% | 83.1% | 85.2% |
| APGD-CE | 66.3% | 77.4% | 72.3% | 64.4% | 67.2% | 66.1% | 71.1% | 70.8% |
| APGD-DLR | 65.6% | 77.6% | 72.0% | 63.0% | 66.0% | 65.3% | 70.5% | 70.6% |
| APGD-T | 65.1% | 77.3% | 71.5% | 62.1% | 65.5% | 64.5% | 69.4% | 69.6% |
| FAB-T | 65.0% | 77.4% | 71.7% | 62.7% | 65.7% | 64.5% | 88.7% | 90.4% |
| SQUARE | 81.2% | 86.2% | 81.7% | 72.0% | 77.1% | 72.2% | 90.2% | 92.1% |
| $\ell_2$ attacks ($\epsilon = 0.5$) | 64.6% | 77.2% | 71.5% | 61.8% | 65.5% | 64.5% | 69.3% | 69.4% |
| PGD-$\ell_1$ | 30.2% | 48.5% | 62.5% | 50.8% | 61.0% | 58.2% | 59.8% | 64.1% |
| FAB-T | 35.0% | 47.2% | 61.3% | 48.3% | 63.8% | 57.7% | 65.5% | 69.3% |
| $\ell_1$ attacks ($\epsilon = 10$) | 27.6% | 45.3% | 60.9% | 43.7% | 60.0% | 56.1% | 57.9% | 59.5% |
| All Attacks | 27.6% | 32.9% | 35.0% | 31.5% | 39.3% | 43.5% | 53.5% | 54.9% |
| Average All Attacks | 60.9% | 59.0% | 57.9% | 53.5% | 57.2% | 57.4% | 71.6% | 73.3% |

Table 13: Effect of the number (n) of specialized robust predictors $M_A$ in PROTECTOR(n) on CIFAR-10. The analysis was performed for an architecture that only utilizes the raw input, and not the Fourier features.

| | PROTECTOR(2) | PROTECTOR(3) |
|---|---|---|
| Clean accuracy | 90.8% | 92.2% |
| APGD $\ell_\infty$ ($\epsilon = 0.03$) | 64.8% | 56.3% |
| APGD $\ell_2$ ($\epsilon = 0.5$) | 68.8% | 69.2% |
| Sparse $\ell_1$ ($\epsilon = 10$) | 55.9% | 52.3% |