# OpenReview forum: "Perturbation Type Categorization for Multiple Adversarial Perturbation Robustness"
_auai.org/UAI/2022/Conference — UAI 2022 Poster_

### Official Review · Reviewer_ReTd · 2022-03-28

**Q2(1) Originality/Novelty:** 3
**Q2(2) Significance/Impact:** 3
**Q2(3) Correctness/Technical Quality:** 3
**Q2(6) Clarity Of Writing:** 2
**Q6 Overall Score:** 6
**Q8 Confidence In Your Score:** 3

**Q1 Summary And Contributions:**

This work seeks adversarial robust classifiers with respect to multiple perturbation types at the same time.
While existing methods tend to perform worse than a robust classifier trained for a specific perturbation, the proposed method leverages the perturbation classifier to categorize which perturbation type input is, and delegates it to a classifier designated to each perturbation.
The empirical evaluation witnesses that the proposed method performs comparably to the specialized classifiers.

**Q2 Assessment Of The Paper:**

More detailed information regarding each of these aspects is given below:

**Q2(4) Quality Of Experiments (Optional):**

3: Good: The experimental evaluation is adequate, and the results convincingly support the main claims.

**Q2(5) Reproducibility:**

4: Excellent: Key resources (e.g., proofs, code, data) are available and key details (e.g., proof sketches, experimental setup) are comprehensively described for competent researchers to confidently and easily reproduce the main results.

**Q3 Main Strengths:**

While most of the existing works on adversarial robustness deal with a specific perturbation, it is practically important to make classifiers robust against multiple perturbations at the same time.
This work provides a practical solution to this important problem by a simple two-step algorithm built on top of a perturbation type classifier.

The theoretical evidence that perturbation types can be classified with high probability is technically interesting in itself.

The proposed method is simple to implement and flexible to incorporate arbitrary state-of-the-art robust classifiers in the second step.

**Q4 Main Weakness:**

The main theorems (Theorems 1 and 2) rely on the assumption that input data are binary and Gaussian distributed, which is restrictive in general, though the experimental results suggest that the proposed method works well anyway.

Although the authors mention the existing methods dealing with multiple perturbation types become inferior to classifiers robust against specific perturbation types (in the second paragraph of the introduction), it is not clear what makes the existing methods inferior and the proposed method overcomes this issue.
I would like to see a little bit of discussion and qualitative comparison of the existing methods and the proposed method.

**Q5 Detailed Comments To The Authors:**

- Though not directly related, the authors may consider citing and discussing Wang et al. (2020), who used conditional training to make a classifier robust against multiple different magnitudes of the accuracy-robustness trade-off.

  Wang et al. (2020) "Once-for-All Adversarial Training: In-Situ Tradeoff between Robustness and Accuracy for Free"

- Just for curiosity, is it straightforward to extend the main theorems to the multi-class setting and $\ell\_p$ perturbations other than $p \in \\{1, \infty\\}$?

- Do we need bold-face for $P\_e$, $\epsilon\_1$, and $\epsilon\_\infty$ in Theorem 2?

- In Sec. 5.1 "Combining perturbation types", the authors "hypothesize that compared to $\ell\_\infty$ adversarial examples, $\ell\_1$ and $\ell\_2$ adversarial examples show similar characteristics." Is it possible to provide any intuition for this in addition to the subsequent numerical evidence?

- In Sec. 5.2, the authors "train $C\_\mathrm{adv}$ over a static dataset", but I do not understand how this dataset is generated even after looking at Sec. E.2. When generating adversarial examples, a targeted classifier needs to be given, which I suppose is fixed in advance.  But is it realistic to have a sufficiently robust targeted classifier from the beginning of the training? Or am I missing something?

- In Sec. 5.3 "Constraining the adversary using random noise", the authors motivate to add random noises to the model input by showing an illustrative example in Figure 2b, which makes a lot of sense. On the other hand, I do not see a direct implication from Theorem 2 motivating the necessity of random noises. Can you elaborate it a little bit more?

**Q7 Justification For Your Score:**

This work solves an important problem to make classifiers robust against multiple perturbation types.
The simple proposed method is based on theoretical evidence, and the empirical performance is good.
On the other hand, the main theorems rely on relatively restrictive assumptions.
Considering the above, I am willing to weakly accept this paper.

**Q9 Complying With Reviewing Instructions:**

1: Yes.

---

### Official Review · Reviewer_Hi14 · 2022-04-12

**Q2(1) Originality/Novelty:** 3
**Q2(2) Significance/Impact:** 3
**Q2(3) Correctness/Technical Quality:** 3
**Q2(6) Clarity Of Writing:** 3
**Q6 Overall Score:** 5
**Q8 Confidence In Your Score:** 3

**Q1 Summary And Contributions:**

This work is motivated by the observation that methods against multiple perturbation types suffer significant trade-offs compared to the ones trained to be robust against a single perturbation type.
Therefore, they theoretically and empirically demonstrate that the perturbation types are distinct and separable. Then they proposed a two-stage pipeline that first categorizes the perturbation type of the input, and adopted the predictor specifically trained against the predicted perturbation type.

**Q2 Assessment Of The Paper:**

More detailed information regarding each of these aspects is given below:

**Q2(4) Quality Of Experiments (Optional):**

3: Good: The experimental evaluation is adequate, and the results convincingly support the main claims.

**Q2(5) Reproducibility:**

3: Good: Key resources (e.g., proofs, code, data) are available and key details (e.g., proofs, experimental setup) are sufficiently well-described for competent researchers to confidently reproduce the main results.

**Q3 Main Strengths:**

The motivation of this paper is strong that a single model against the union of multiple perturbation types still suffers significant trade-offs compared to the ones specifically trained to be robust against a single perturbation type.

The theoretical analysis and empirical study are insightful. The claim that perturbation types are distinct and separable is well supported.

It is interesting to see the natural tension for any adversary trying to fool the proposed model.

The paper is well organized and easy to read.

**Q4 Main Weakness:**

My main concern is that it seems we should know about the perturbation types in advance such that we can build the perturbation classifier. Besides, the robust model is also trained on a specific dataset where certain types of attacks were combined under the same label.

It requires a lot of prior knowledge and may not be feasible in real-world applications.

**Q5 Detailed Comments To The Authors:**

There should be punctuation at the end of the equations.

**Q7 Justification For Your Score:**

This work introduced a new problem and proposed an effective method. The theoretical analysis and empirical study are sufficient. However, it requires some prior knowledge, which may not be feasible in real-world applications. Thus I arrived at 5.

**Q9 Complying With Reviewing Instructions:**

1: Yes.

---

### Official Review · Reviewer_9mCf · 2022-04-12

**Q2(1) Originality/Novelty:** 2
**Q2(2) Significance/Impact:** 1
**Q2(3) Correctness/Technical Quality:** 1
**Q2(6) Clarity Of Writing:** 2
**Q6 Overall Score:** 3
**Q8 Confidence In Your Score:** 4

**Q1 Summary And Contributions:**

This manuscript proposes to implement a classifier to predict the type of adversarial attacks. This prediction would then activate a dedicated predictive model that has been trained explicitly on this kind of adversarial attack. From this two-staged approach one would expect better robustness performance, in comparison with the traditional way of training one model with various adversarial attacks.

**Q2 Assessment Of The Paper:**

More detailed information regarding each of these aspects is given below:

**Q2(4) Quality Of Experiments (Optional):**

2: Fair: The experimental evaluation is weak: important baselines are missing, or the results do not adequately support the main claims.

**Q2(5) Reproducibility:**

2: Fair: Key resources (e.g., proofs, code, data) are unavailable but key details (e.g., proof sketches, experimental setup) are sufficiently well-described for an expert to confidently reproduce the main results.

**Q3 Main Strengths:**

The new idea of training a classifier for the adversarial types seems interesting and worth investigation.

**Q4 Main Weakness:**

Pleas refer to Q5.

**Q5 Detailed Comments To The Authors:**

It is an intriguing and ambitious idea to classify the adversarial attacks. However, this manuscript does not yet provide convincing materials that would support the claim.
- First, the adversarial attacks, especially the gradient based ones, are known to be subtle and imperceivable. The qualitative examples provided in Fig. 2 are not expected to generalize. Even if they would, the scenario described in b) already suggests that comparing the norm of the perturbation vector would provide enough information for the type classification. Classifying non-gradient based adversarial attacks such as blurring and speckle noises etc. is not a challenging task.
- Second, the separability theorem in section 3.1. assumes that there's one input feature that correlates with the target "strongly", which can not be guaranteed in practice. Furthermore, A type classifier of only two classes would be really limited in practice. Maybe for future work the authors want to combine multiple such binary classifiers into a binary search.
- A couple of minor issues in terms of notation and terminology: in section 3.2., for instance, I would think that the classifier M is the model for the actual predictive task but in Theorem 1 it seems to be re-defined as the binary classifier for the attack type. In addition, I don't think "Gaussian Classifier" is a well known term in the community. Is it the Naive Bayes or LDA?

**Q7 Justification For Your Score:**

I find the idea in general very ambitious and worth further analyzing. However, the paper as it is does not yet provide a relevant impact and would require a large scale revision.

**Q9 Complying With Reviewing Instructions:**

1: Yes.

---

### Decision · Program_Chairs · 2022-05-15

**Decision:**

Accept (Poster)

**Comment:**

Meta Review: The paper studies the properties of adversarial attacks and proposes to categorize the attacks to better defend them, which is very interesting. Overall, the paper is well written and organised. Experiments show the effectiveness of the proposed method. The meta-reviewer is happy to recommend acceptance as we believe utilizing the properties of adversarial attacks, e.g., via modelling adversarial noise, is promising for adversarial defence.